# ENTROPE: ENTROPY-GUIDED DYNAMIC PATCH ENCODER FOR TIME SERIES FORECASTING

## ABSTRACT

Transformer-based models have significantly advanced time series forecasting, with patch-based input strategies offering efficiency and improved long-horizon modeling. Yet, existing approaches rely on *temporally-agnostic* patch construction, where arbitrary starting positions and fixed lengths fracture temporal coherence by splitting natural transitions across boundaries. This naive segmentation often disrupts short-term dependencies and weakens representation learning. In response, we propose **EntroPE** (**Entro**py-Guided Dynamic **P**atch **E**ncoder), a novel, temporally informed framework that dynamically detects transition points via conditional entropy and dynamically places patch boundaries. This preserves temporal structure while retaining the computational benefits of patching. EntroPE consists of two key modules, namely an **Entropy-based Dynamic Patcher (EDP)** that applies information-theoretic criteria to locate natural temporal shifts and determine patch boundaries, and an **Adaptive Patch Encoder (APE)** that employs pooling and cross-attention to capture intra-patch dependencies and produce fixed-size latent representations. These embeddings are then processed by a global transformer to model inter-patch dynamics. Experiments across long-term forecasting benchmarks demonstrate that EntroPE improves both accuracy and efficiency, establishing entropy-guided dynamic patching as a promising new paradigm for time series modeling.

## 1 INTRODUCTION

Time series analysis is fundamental to numerous critical applications, including electricity load forecasting (Gasparin et al., 2022), financial market prediction (Fischer & Krauss, 2018), healthcare monitoring (Reis & Mandl, 2003), and environmental surveillance (Smith, 1989). However, accurately modeling time series data remains challenging due to inherent noise, complex temporal dependencies, and irregular patterns spanning multiple time horizons (Lim & Zohren, 2021; Torres et al., 2021). The transformer architecture (Vaswani et al., 2017) has revolutionized sequence modeling across domains through self-attention mechanisms, leading to significant advances in time series forecasting. Despite their promise in learning long-term temporal relationships, time series transformers remain suboptimal due to unique complexities such as non-stationarity, seasonality, varying temporal granularities (Wen et al., 2023; Zeng et al., 2023).

Recent advances in time series transformers include Informer (Zhou et al., 2021), Autoformer (Wu et al., 2021), and FEDformer (Zhou et al., 2022), which address long-sequence modeling challenges through sparse attention mechanisms, decomposition techniques, and improved positional encodings while operating on point-wise inputs. Triformer (Cirstea et al., 2022) and PatchTST (Nie et al., 2023) introduced a patch-based input representation, dividing raw sequences into fixed-length patches and achieving substantial improvements in computational efficiency and long-term forecasting performance. This demonstrated the critical impact of input tokenization strategies on transformer performance in time series. However, subsequent approaches have predominantly adopted similar temporally-agnostic patching schemes, which may not fully capture the temporal coherence and statistical properties inherent in time series data.

Such temporally-agnostic patching introduces critical technical challenges. Fixed-length patching creates inconsistent input representations between training and inference phases. During training, time series samples from different time periods result in patches that begin at various temporal po-

sitions relative to underlying patterns (e.g., a patch might start mid-trend in one sample but at a trend beginning in another). However, during inference, patches are extracted from predetermined positions in the input sequence, creating a systematic distribution shift where the model encounters patch configurations it rarely or never saw during training. Further, predetermined patch boundaries arbitrarily segment coherent temporal structures without regard for underlying dynamics (see Fig. 1(a)). For example, a gradual trend change or seasonal transition may be split across multiple patches, breaking the natural temporal dependencies that are crucial for accurate pattern recognition and forecasting.

These challenges directly impact model performance through two mechanisms. Fragmented temporal patterns lead to incomplete intra-patch representations, where semantically related time points are separated across different patches and processed independently. Additionally, the train-inference mismatch reduces the model's ability to generalize, as it must extrapolate to patch configurations with different statistical properties than those encountered during training. This is particularly detrimental for capturing rapid transitions and fine-grained temporal dynamics that require consistent temporal context.

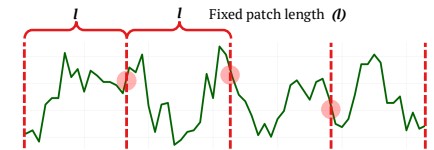

(a) **Temporally-agnostic Patching:** Patch boundaries are placed without regard to temporal coherence, leading to heterogeneous fragmentation of patterns.

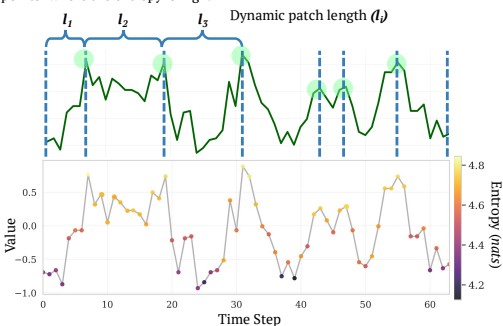

(b) **Temporally-informed Patching:** Patch boundaries lie at natural transition points where the entropy is high.

Unlike prior work that treats patching as a static preprocessing step, we propose **EntroPE**, an **En**tro**py**-Guided Dynamic **P**atch **E**ncoder, which integrates dynamic boundary detection as a core architectural component. Our approach employs entropy-based criteria to identify natural temporal transition points, ensuring patch boundaries align with the underlying temporal structure rather than arbitrary positions (see Fig. 1(b)). We then employ adaptive encoding mechanisms to process these variable-length patches while preserv-

Figure 1: **(a)** Temporally-agnostic, and **(b)** Temporally-informed (proposed) patching with calculated entropies for ETTh1 sample.

ing intra-patch dependencies. This temporally-informed approach addresses both the train-inference mismatch and boundary fragmentation issues inherent in temporally-agnostic patching methods.

The main contributions of this paper are summarized as follows:

- We introduce the first information-theoretic approach to patching for time series forecasting, where conditional entropy is used to place boundaries that respect temporal causality and predictive difficulty.

- We design an adaptive encoder that converts variable-length patches into fixed representations via pooling and cross-attention, enabling efficient batch computation.

- We demonstrate consistent gains across diverse benchmarks, showing that entropy-guided dynamic patching is a practical and generalizable direction for time series transformers.

## 2 RELATED WORK

**Time Series Transformers and Long-Term Forecasting.** The transformer architecture has fundamentally transformed sequence modeling across domains, with time series forecasting being no exception. Early adaptations like Informer (Zhou et al., 2021) addressed the quadratic complexity of self-attention in long sequences through ProbSparse attention mechanisms, while Autoformer (Wu et al., 2021) and FEDformer (Zhou et al., 2022) introduced decomposition-based approaches and frequency domain modeling to capture complex temporal dependencies. These foundational works established transformers as viable architectures for time series tasks but operated on point-wise inputs, leading to computational challenges and a limited ability to capture local temporal patterns efficiently. The fundamental challenge these approaches faced was the trade-off between sequence length and computational tractability. Point-wise processing, while preserving fine-grained tempo-

ral information, resulted in prohibitively long input sequences for practical applications, particularly in long-term forecasting scenarios where historical contexts spanning hundreds or thousands of time steps are crucial for accurate predictions.

**Patch-Based Input Strategies.** Inspired by vision transformers (Dosovitskiy et al., 2021), patch-based input strategies have been widely adopted in time series forecasting. Triformer (Cirstea et al., 2022) and PatchTST (Nie et al., 2023) segment time series into fixed-length patches that serve as input tokens. This design achieves a quadratic reduction in sequence length while preserving local semantic information through channel-independent processing, demonstrating that tokenization strategies can significantly improve both computational efficiency and forecasting accuracy. Building on this foundation, Crossformer (Zhang & Yan, 2023) introduced hierarchical mechanisms tailored for modeling both intra-patch and inter-patch dependencies. CARD (Xue et al., 2024) extended patching to multivariate forecasting, while patch-independence approaches (Lee et al., 2024) revealed superior representations in self-supervised settings which further highlights the importance of patch-based input strategies for time series modeling. Beyond Transformer architectures, xPatch (Stitsyuk & Choi, 2025) combined dual-flow MLP–CNN designs with exponential decomposition, whereas Pathformer (Chen et al., 2024) and PatchMLP (Tang & Zhang, 2025) leveraged hierarchical processing to capture multi-scale relationships.

However, fixed-length patching introduces key limitations. Uniform segmentation can fragment coherent patterns (e.g., seasonal transitions, trend shifts), hindering the learning of complete temporal dependencies. It also induces a train-inference mismatch, where during training similar windows may begin at arbitrary positions within underlying patterns, while inference applies fixed boundaries that force the model to generalize to configurations rarely observed in training. These limitations can result in poor generalization, weakened representation learning, and the loss of short-term dependencies. In response, MSPatch (Cao et al., 2025) introduces multi-scale patch segmentation to capture both short- and long-term dependencies, mitigating the fragmentation of coherent temporal patterns. HDMixer (Huang et al., 2024) proposes a Length-Extendable Patcher (LEP) module that adjusts patch lengths via bi-linear interpolation to enrich boundary information, though it remains constrained by predetermined patch counts and may introduce interpolation-induced uncertainty. Beyond patch resizing, foundation models have explored alternative input constructions such as MOIRAI (Woo et al., 2024) leverages frequency-based patch sizing to preserve periodic structures, while Chronos (Ansari et al., 2024) bypasses patching entirely by treating individual time points as tokens, albeit at a substantially higher computational cost.

While recent works have advanced patch construction strategies, significant gaps remain. Existing adaptive methods often overlook causal structure and temporal coherence. This prevents them from capturing optimal boundaries guided by temporal uncertainty and predictive difficulty. Given the highly dynamic nature of time series across domains, a principled and fully adaptive approach is required. Moreover, efficient training capabilities are required, particularly due to the variable number of patches in each sample. In contrast, dynamic patching schemes in other domains, such as natural language processing, have shown superior performance and efficiency. For instance, the Byte Latent Transformer (Pagnoni et al., 2025) employs byte level inputs instead of traditional word tokenization, and dynamically decide patches as a part of the architeture. Motivated by these advances, our work introduces entropy-guided dynamic boundary detection that respects temporal causality and predictive uncertainty, coupled with adaptive encoding mechanisms specifically designed to preserve temporal dependencies within variable-length patches.

## 3 METHODOLOGY

### 3.1 PROBLEM FORMULATION

Given a multivariate time series $X = [x_1, x_2, \ldots, x_C] \in \mathbb{R}^{C \times L}$ with $C$ channels and look-back length $L$, the goal is to predict $\hat{Y} \in \mathbb{R}^{C \times T}$ over forecast horizon $T$. Following the channel-independence principle from PatchTST (Nie et al., 2023), we deliberately disregard cross-channel dependencies and apply a shared function $f_\phi : \mathbb{R}^L \to \mathbb{R}^T$ (with shared parameters $\phi$) to each channel independently: $\hat{y}_c = f_\phi(x_c), \quad c = 1, 2, \ldots, C.$

This design choice isolates the temporal modeling capabilities from cross-variate relationships, allowing us to focus purely on the effectiveness of our dynamic patching mechanism within each

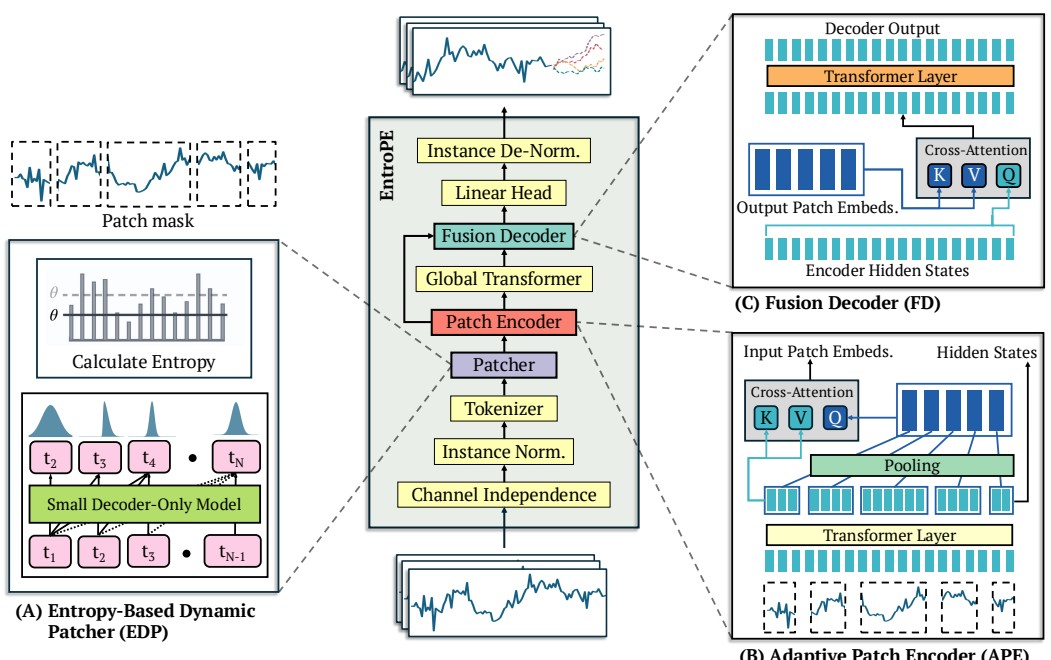

Figure 2: Comprehensive architecture of EntroPE. The model processes input through: (A) **Entropy-Based Dynamic Patcher** - A small causal transformer calculates entropy at each time point to identify boundaries where predictive uncertainty is high; (B) **Adaptive Patch Encoder** - Cross-attention layers aggregate intra-patch dependencies into fixed-size global embeddings; (C) **Fusion Decoder** - Cross-attention combines global patch context with local encoder hidden states for accurate forecasting.

univariate series. The multivariate prediction is constructed by stacking channel-wise results:

$$\hat{Y} = [\hat{y}_1, \hat{y}_2, \ldots, \hat{y}_C] \in \mathbb{R}^{C \times T}$$

## 3.2 FORWARD PROCESS

The forward process of EntroPE (see Fig. 2) transforms raw multivariate time series into forecasts through a sequence of interconnected components. The raw input multivariate time series $X \in \mathbb{R}^{B \times C \times L}$, where $B$, $C$, and $L$ represent batch size, number of channels (variables), and sequence length respectively, is first processed following the channel independence principle, converting it to $X^{ci} \in \mathbb{R}^{(B \cdot C) \times L}$ for independent processing of each variable. As the next step, to mitigate the statistical difference between training and testing distributions, we integrate the Reversible Instance Normalization layer (Kim et al., 2021) which outputs $X^{Norm.} \in \mathbb{R}^{(B \cdot C) \times L}$.

Following the Chronos tokenization approach (Ansari et al., 2024) (Appendix A.7), we discretize the continuous input sequence into uniform bins based on a fixed-size vocabulary $V$, producing $X^{tok}$. The tokenized sequence $X^{tok}$ is then processed by the **EDP** (explained in Sec. 3.3), which generates a patch mask $M \in \mathbb{R}^{B \cdot C \times L}$ indicating optimal patch boundary locations. Using this patch mask, the original continuous input $X^{ci}$ is segmented into variable-length patches. These patches are fed into the **APE** (explained in Sec. 3.4), which generates patch embeddings while aggregating intra-patch dependencies.

The encoder produces two key outputs, namely, (1) patch embeddings $P^{in} \in \mathbb{R}^{B \cdot C \times \hat{p} \times d_p}$, which serve as input embeddings for the global transformer, and (2) encoder hidden states $H^{enc} \in \mathbb{R}^{B \cdot C \times L \times d_t}$ that capture temporal information at the original sequence resolution for subsequent decoding head. Here, $\hat{p}$ is not fixed due to the dynamic nature of the patching, $d_t$ and $d_p$ are time point-wise embedding and patch embedding dimensions, respectively.

The patch embeddings $P^{in}$ are processed by the **Global Transformer**, which learns inter-patch long-term dependencies and outputs enriched patch embeddings $P^{out} \in \mathbb{R}^{B \cdot C \times \hat{p} \times d_p}$. Finally,

the **Fusion Decoder (FD)** combines the global transformer outputs ($P^{out}$) with the encoder hidden states ($H^{enc}$) through cross-attention, flattens the enriched representations, and projects them through a linear head and Instance De-Normalization layer to generate predictions for the future $T$ time steps, producing the final forecast $\hat{Y} \in \mathbb{R}^{B \times C \times T}$.

## 3.3 ENTROPY-BASED DYNAMIC PATCHER (EDP)

As shown in Fig. 2(A), our dynamic patch boundary detection leverages a lightweight causal transformer to identify temporal transitions based on predictive uncertainty. The process operates in two independent phases, namely model pre-training for temporal pattern learning and entropy-guided patch boundary detection with frozen weights.

**Entropy Model Pre-training.** Drawing inspiration from the decoder-only transformer architecture and cross-entropy training demonstrated in Chronos, we adopt the GPT-2 paradigm (Radford et al., 2019) to train a compact transformer for next-token prediction on quantized time series tokens. Our lightweight architecture comprises 8 embedding dimensions (shared input and head), 2 transformer layers, 4 attention heads, totaling 3,648 learnable parameters. The model performs autoregressive prediction at time $\{t_2, t_3, \ldots, t_L\}$ from input sequence at time $\{t_1, t_2, \ldots, t_{L-1}\}$ using standard cross-entropy loss.

This streamlined design efficiently captures temporal dynamics while maintaining computational tractability for entropy estimation. Crucially, our objective is not to achieve superior forecasting performance, but rather to align the causal transformer for measuring predictive uncertainty across the discrete vocabulary. This architectural approach enables identification of natural transition points in raw time series sequences, where predictive uncertainty that quantified through entropy over the output next-token distribution reaches local maxima.

**Entropy-Guided Patch Boundary Detection.** Upon training convergence, we freeze model weights and compute entropy for each time position. For position $t$, we calculate the Shannon entropy of the next-token prediction distribution:

$$H(x_t) = - \sum_{v \in \mathcal{V}} p_\theta(x_{t+1} = v | x_{\leq t}) \log p_\theta(x_{t+1} = v | x_{\leq t}) \tag{1}$$

Following Pagnoni et al. (2025), we implement dual-threshold boundary detection combining global and relative uncertainty measures. A patch boundary is placed at position $t$ when three conditions are met simultaneously:

$$H(x_t) > \theta \quad \text{(global entropy threshold)} \tag{2}$$
$$H(x_t) - H(x_{t-1}) > \gamma \quad \text{(relative entropy increase)} \tag{3}$$

This mechanism ensures patches begin only at positions with both high absolute uncertainty and significant uncertainty increases, avoiding fragmentation from minor entropy fluctuations while capturing meaningful temporal transitions. EDP outputs patch mask $M$, which indicates the non-overlapping patch segmentation for further processing.

## 3.4 ADAPTIVE PATCH ENCODER (APE)

The EDP produces variable-length patches where both patch count $\hat{p}$ and individual lengths vary across samples (patch mask $M$). As shown in Fig. 2(B), the APE converts these heterogeneous patches into fixed-size representations suitable for transformer processing while preserving temporal information. When handling variable patches, we ensure batch processing compatibility by padding sequences to the maximum patch count within each batch and applying attention masks to handle variable patch lengths during cross-attention operations.

**Architecture Design.** Our encoder employs a two-stage approach. First initial dimensionality reduction via pooling (from time-point embeddings to initial patch embeddings), followed by iterative cross-attention refinement. This design is inspired by the Perceiver architecture (Jaegle et al., 2021) and Byte Latent Transformer (Pagnoni et al., 2025), but specialized for temporal patch aggregation.

**Initial Patch Embedding.** We first obtain the time-point embedding $h_0$ for the entire sequence with length $L$ through an Embedding Layer $E \in \mathbb{R}^{V \times d_t}$ over the same Vocabulary and Tokenizer used

in the Entropy Model. For each variable-length patch $p_j$, we generate initial patch representations via max pooling: $P_{0,j} = \text{MaxPool}(E_f(p_j))$, where $E_f$ represents the time-point embedding function. Max pooling captures salient temporal features while providing translation invariance within patches.

**Cross-Attention Refinement.** We refine patch representations using cross-attention, where patch embeddings serve as queries and time-point embeddings act as keys and values. For layer $n$:

$$P_n = P_{n-1} + W_o \left( \text{softmax}\left(\frac{QK^T}{\sqrt{d_k}}\right) V \right), \tag{4}$$

with attention components:

$$Q_j = W_q(P_{n-1,j}) \quad \text{(patch embeddings as queries)}, \tag{5}$$
$$K_i = W_k(h_{n-1,i}), \quad V_i = W_v(h_{n-1,i}) \quad \text{(time-point embeddings as keys/values)}, \tag{6}$$

where $W_q$, $W_k$, $W_v$, and $W_o$ are learnable projection matrices corresponding to queries, keys, values, and output transformations, respectively. For $N$-layer architectures with $N > 1$, time-point embeddings are iteratively refined through stacked transformer layers as $h_n = \text{APETransformer}(h_{n-1})$, with cross-attention mechanisms operating between consecutive layers. Here, $h_n$ is the time point embedding sequence at layer $n$.

To ensure that self-attention within each APETransformer operates exclusively within patch boundaries, we employ a tailored attention mask that accommodates the variable patch lengths throughout the sequence. This masking strategy prevents information leakage across patch boundaries while preserving intra-patch temporal dependencies.

The encoder outputs fixed-size patch embeddings $P^{in} = P_N \in \mathbb{R}^{B \cdot C \times \hat{p} \times d_p}$ for global transformer processing and preserves encoder hidden states $H^{enc} = h_N \in \mathbb{R}^{B \cdot C \times L \times d_t}$ for temporal information in the decoding stage.

## 3.5 GLOBAL TRANSFORMER & FUSION DECODER

The final components of the EntroPE process are the fixed-size patch embeddings to learn long-term dependencies and generate forecasts.

**Global Transformer.** The patch embeddings $P^{in}$ from the adaptive encoder are processed through a standard transformer architecture to capture long-range temporal relationships between patches. Since intra-patch dependencies are already modeled by the encoder, the global transformer focuses exclusively on inter-patch interactions. We employ bidirectional attention (non-causal) to enable comprehensive context modeling across the entire sequence: $P^{out} = \text{GlobalTransformer}(P^{in})$.

**Fusion Decoder (FD).** A key challenge in dynamic patching is that the number of patches $\hat{p}$ varies across samples, making direct forecasting from flattened patch representations problematic. To address this, we employ a cross-attention mechanism that fuses global patch context with fine-grained temporal information preserved from the encoder (see Fig. 2(C)). The decoder reverses the attention configuration used in the adaptive encoder. Here, encoder hidden states $H^{enc}$ serve as queries, while global patch embeddings act as keys and values:

$$Q_i = W_q(H_i^{enc}) \quad \text{(time-point queries)}, \tag{7}$$
$$K_j = W_k(P_j^{out}), \quad V_j = W_v(P_j^{out}) \quad \text{(patch keys/values)} \tag{8}$$

The cross-attention operation enriches time-point representations with global context:

$$H^{dec} = H^{enc} + W_o \left( \text{softmax}\left(\frac{QK^T}{\sqrt{d_k}}\right) V \right). \tag{9}$$

This mechanism enables knowledge transfer from high-level patch representations back to detailed time-point embeddings, ensuring preservation of both local temporal patterns and global contextual dependencies. Optionally, similar to APE, the iterative cross-attention refinement can be configured through alternating transformer layers and cross-attention layers, creating a structured information exchange between different representational levels.

**Forecasting Head.** The enriched representations $H^{dec}$ are projected to generate future predictions. We apply flattening and linear projection with Instance De-Normalization: $\hat{Y} = $ InstanceDeNorm$\left(W_{proj}(\text{Flatten}(H^{dec}))\right)$, where $\hat{Y} \in \mathbb{R}^{B \times C \times T}$ represents the forecasted values for the next $T$ time steps, maintaining the original multivariate structure. The model is trained using Mean Squared Error (MSE) loss $L_{MSE}(Y, \hat{Y})$ between predicted and ground truth future values.

## 4 EXPERIMENTS

**Datasets.** We evaluate EntroPE on seven widely-used long-term multivariate forecasting benchmarks: ETT family (ETTh1, ETTh2, ETTm1, ETTm2), Weather, Electricity, and Exchange Rate datasets, covering diverse domains and temporal characteristics (detailed descriptions in Appendix Tab. 4).

**Baselines.** We benchmark against 14 state-of-the-art deep forecasting models from 2021–2025, organized by architecture. (1) Transformer-based models: iTransformer (Liu et al., 2024b), PatchTST (Nie et al., 2023), FEDformer (Zhou et al., 2022), Autoformer (Wu et al., 2021); (2) CNN-based model: TimesNet (Wu et al., 2023); (3) MLP-based models: TimeMixer (Wang et al., 2024), HD-Mixer (Huang et al., 2024), DLinear (Zeng et al., 2023); (4) Kernel-attention model: TimeKAN (Huang et al., 2025b); (5) Foundation and lightweight models: Time-FFM (Liu et al., 2024a), Time-Base (Huang et al., 2025a); (6) LLM-aligned and cross-modal models: LangTime (Niu et al., 2025), CALF (Liu et al., 2025); (7) Filter-based model: FilterTS (Wang et al., 2025). (Further details on baselines are discussed in Appendix A.2.)

**Experimental Settings.** Following standard practices in time series forecasting literature, we evaluate performance using Mean Squared Error (MSE) and Mean Absolute Error (MAE) metrics across multiple prediction horizons $T = \{96, 192, 336, 720\}$ and fixed look-back window $L = 96$. Adopting the same experiemnt setup as Huang et al. (2025b), baseline results up to 2024 are taken from TimeKAN, while other methods are re-evaluated using their official implementations. EntroPE experiemnts are conductd in 5 random seeds and we report the best results. (see Appendix A.4.)

### 4.1 MAIN RESULTS

Table 1 summarizes average performance across all prediction horizons. EntroPE matches or surpasses, in many cases, recent variable-mixing, Transformer-based, CNN-based, MLP-based, foundation, lightweight, LLM-aligned, filter-based, and KAN-based models under the standard 96-step look-back with horizons up to 720. It retains PatchTST's overall architecture but introduces two key innovations namely, (i) entropy-driven dynamic patching and (ii) adaptive embeddings. Relative to PatchTST, EntroPE achieves notable accuracy gains of approximately 20% on ETTh1, 15% on Electricity, and about 10% on average, while also reducing token count and improving efficiency by leveraging channel independence and non-overlapping patches. Beyond PatchTST, EntroPE consistently outperforms HDMixer's length-extendable patching and improves upon TimeBase, though the latter remains highly efficient. On the Electricity dataset, iTransformer continues to offer the strongest performance-efficiency tradeoff given its channel-wise self-attention across 321 and 862 variables, respectively. CALF and LangTime also remain competitive, though their large model sizes (CALF has 18 million, LangTime has 500 million while EntroPE has 0.1 million range parameters) impose substantial costs. Across other benchmarks, EntroPE delivers robust improvements, underscoring the value of entropy-guided temporal segmentation for multivariate forecasting.

### 4.2 ABLATION STUDIES

**Model component analysis.** To systematically evaluate each component's contribution, we conduct comprehensive ablation studies across four datasets (ETTh1, ETTh2, ETTm1, Weather) using prediction horizons of 336 and 720 time steps, measuring performance via MSE. We progressively remove components from our full EntroPE architecture to create four configurations: (1) EntroPE (Full): Dynamic patching + Adaptive encoder + Fusion decoder; (2) EntroPE - Dynamic: Static patching + Adaptive encoder + Fusion decoder; (3) EntroPE - (Dynamic Patching + Adaptive Encoder): Static patching + Max pooling + Fusion decoder; (4) EntroPE - (Dynamic Patching + Adaptive Encoder + Fusion Decoder): Static patching + Max pooling + Flattened output.

Table 1: Multivariate time series forecasting results on benchmark datasets. Results are averaged across prediction horizons $T = \{96, 192, 336, 720\}$ with fixed input length $L = 96$ for all datasets. Best results are highlighted in **red** and second-best in **blue**. A dash (-) indicates that the configuration was not found in the original implementation. (Full table: Appendix A.5)

| Models | ETTh1 MSE | ETTh1 MAE | ETTh2 MSE | ETTh2 MAE | ETTm1 MSE | ETTm1 MAE | ETTm2 MSE | ETTm2 MAE | Weather MSE | Weather MAE | Electricity MSE | Electricity MAE | Exchange MSE | Exchange MAE |
|---|---|---|---|---|---|---|---|---|---|---|---|---|---|---|
| Autoformer [2021] | 0.496 | 0.487 | 0.450 | 0.459 | 0.588 | 0.517 | 0.327 | 0.371 | 0.338 | 0.382 | 0.227 | 0.338 | 0.613 | 0.539 |
| FEDformer [2022] | 0.498 | 0.484 | 0.437 | 0.449 | 0.448 | 0.452 | 0.305 | 0.349 | 0.309 | 0.360 | 0.214 | 0.327 | 0.519 | 0.429 |
| DLinear [2023] | 0.461 | 0.457 | 0.563 | 0.519 | 0.404 | 0.408 | 0.354 | 0.402 | 0.265 | 0.315 | 0.225 | 0.319 | 0.354 | 0.414 |
| TimesNet [2023] | 0.495 | 0.450 | 0.414 | 0.427 | 0.400 | 0.406 | 0.291 | 0.333 | 0.251 | 0.294 | 0.193 | 0.304 | 0.416 | 0.443 |
| PatchTST [2023] | 0.516 | 0.484 | 0.391 | 0.411 | 0.406 | 0.407 | 0.290 | 0.334 | 0.265 | 0.285 | 0.216 | 0.318 | 0.367 | 0.404 |
| Time-FFM [2024] | 0.442 | 0.434 | 0.382 | 0.406 | 0.399 | 0.402 | 0.286 | 0.332 | 0.270 | 0.288 | 0.216 | 0.299 | **0.338** | **0.391** |
| HDMixer [2024] | 0.448 | 0.437 | 0.384 | 0.407 | 0.396 | 0.402 | 0.286 | 0.331 | 0.253 | 0.285 | 0.205 | 0.295 | - | - |
| iTransformer [2024] | 0.454 | 0.447 | 0.383 | 0.407 | 0.407 | 0.410 | 0.288 | 0.332 | 0.258 | 0.278 | **0.178** | **0.270** | 0.360 | 0.403 |
| TimeMixer [2024] | 0.459 | 0.444 | 0.390 | 0.409 | 0.382 | 0.397 | **0.279** | 0.324 | 0.245 | 0.276 | 0.182 | 0.272 | 0.378 | 0.410 |
| TimeBase [2025] | 0.463 | 0.429 | 0.409 | 0.425 | 0.431 | 0.420 | 0.290 | 0.332 | 0.252 | 0.279 | 0.227 | 0.296 | - | - |
| LangTime [2025] | 0.437 | **0.425** | 0.375 | **0.392** | 0.397 | 0.392 | 0.284 | **0.321** | 0.252 | 0.273 | 0.201 | 0.285 | 0.361 | 0.400 |
| TimeKAN [2025] | **0.418** | 0.427 | 0.391 | 0.410 | **0.380** | 0.398 | 0.285 | 0.331 | **0.244** | **0.273** | 0.197 | 0.286 | - | - |
| FilterTS [2025] | 0.440 | 0.432 | 0.375 | 0.399 | 0.386 | 0.397 | **0.279** | 0.323 | 0.253 | 0.280 | 0.184 | 0.275 | 0.355 | 0.400 |
| CALF [2025] | 0.441 | 0.435 | **0.372** | 0.395 | 0.396 | **0.391** | 0.280 | **0.321** | 0.250 | 0.274 | **0.177** | **0.266** | - | - |
| **EntroPE [2025]** | **0.416** | **0.425** | **0.366** | **0.387** | **0.378** | **0.391** | 0.286 | 0.335 | **0.242** | **0.273** | 0.182 | 0.271 | **0.331** | **0.386** |

The full EntroPE architecture achieves the best performance across all datasets and horizons, while systematic component removal leads to progressive performance degradation (Table 2). The EntroPE - Dynamic Patching configuration yields second-best results, demonstrating the significant contribution of dynamic boundary detection. Further removing the adaptive encoder (EntroPE - (Dynamic Patching + Adaptive Encoder)) shows additional degradation, validating our cross-attention-based encoder design. The worst performance occurs when all three components are removed, producing only static patching with basic pooling and flattened output. This systematic performance decline confirms that each architectural choice addresses specific modeling challenges in time series forecasting.

Table 2: Ablation study showing component importance and dynamic vs. static patching comparison. Best results are highlighted in **red** and second-best in **blue**.

| Dataset | T | Full Dynamic | -EDP Static(1) | -EDP Static(8) | -EDP Static(16) | -EDP -APE Pool + FD | -EDP -APE -FD Pool + Flat | Baselines PatchTST | Baselines iTrans. |
|---|---|---|---|---|---|---|---|---|---|
| ETTh1 | 336 | **0.429** | **0.425** | 0.441 | 0.444 | 0.438 | 0.519 | 0.501 | 0.487 |
| ETTh1 | 720 | **0.439** | **0.460** | 0.469 | 0.477 | 0.461 | 0.527 | 0.503 | 0.500 |
| ETTh2 | 336 | **0.355** | 0.428 | 0.435 | 0.439 | 0.439 | 0.462 | **0.427** | 0.428 |
| ETTh2 | 720 | **0.397** | 0.448 | 0.470 | 0.463 | 0.466 | 0.456 | 0.436 | **0.427** |
| ETTm1 | 336 | **0.393** | **0.401** | 0.409 | 0.415 | 0.402 | 0.427 | 0.421 | 0.426 |
| ETTm1 | 720 | **0.445** | **0.451** | 0.459 | 0.452 | 0.457 | 0.483 | 0.462 | 0.491 |
| Weather | 336 | **0.258** | **0.261** | 0.265 | 0.270 | 0.262 | 0.262 | 0.284 | 0.278 |
| Weather | 720 | **0.341** | **0.343** | 0.353 | 0.359 | 0.347 | 0.346 | 0.356 | 0.358 |

**Dynamic vs. Static Patching.** In addition to the component ablations, we compare EntroPE with dynamic patching against three static (fixed-length, non-overlapping) patching schemes. Experiments are conducted on ETTh1, ETTm1, and Weather datasets with a fixed input length of 96 and forecasting horizons of 336 and 720. Dynamic patching achieves the lowest MSE in most settings (Table 2, while single-token input (patch length of 1) remains competitive, notably without leveraging the Dynamic Patching scheme. However, the computational cost of this setting is substantially higher, as the self-attention mechanism must attend to every time step individually. Figure 3 illustrates the efficiency advantage of dynamic patching over static alternatives under different configurations.

**Threshold Sensitivity.** We further investigate the effect of the entropy threshold on EntroPE's performance using the ETTh1 and Weather datasets, which represent small and medium-scale benchmarks. As shown in Figure 4, the MSE values for the 96→336 forecasting setting remain stable across relative threshold values $\gamma$ in the range 0.15-0.55, indicating robustness to threshold selection. At the same time, training time and patch construction reveal that the threshold effectively acts as a control knob for computational complexity: lower thresholds yield more patches (higher cost), while higher thresholds reduce the number of patches and speed up (31% in Weather and 25% in ETTh1) training, with minimal variation in predictive accuracy.

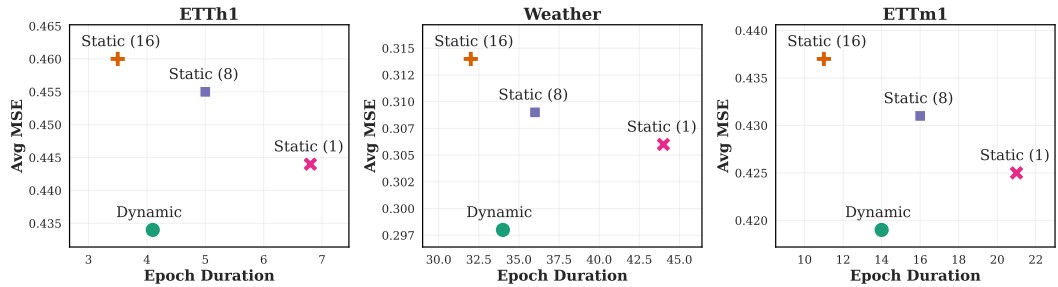

Figure 3: Ablation study for Dynamic and Static(*patch_length*) patching with single epoch duration.

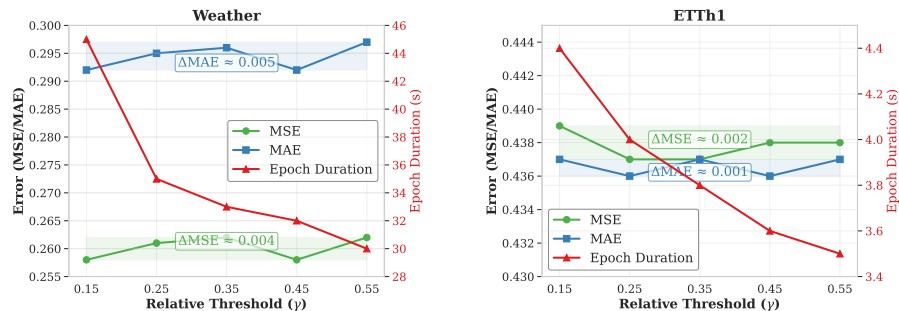

Figure 4: Ablation study for threshold sensitivity with epoch duration.

### 4.3 EFFICIENCY ANALYSIS

As shown in Figures 3 and 4, our model adapts to different patching schemes, directly affecting computational efficiency. In the dynamic setting, the model maintains superior predictive performance while operating within an optimal computation budget. We evaluate efficiency on the ETTm1 dataset (96→96) by comparing recent models in terms of Multiply-Accumulate Operations (MACs), Mean Squared Error (MSE), and the number of trainable parameters. Figure 5 demonstrates that our model achieves a more favorable performance–efficiency trade-off, with bubble size proportional to the parameter count. Notably, while TimeMixer and TimeKAN occupy a nearby region in the trade-off

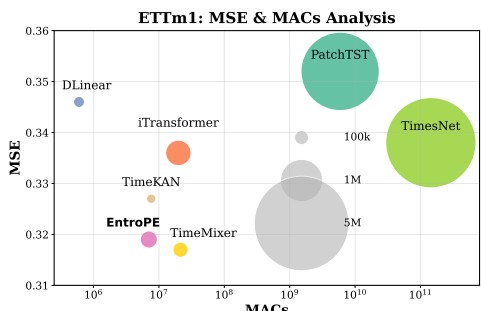

Figure 5: EntroPE efficiency analysis.

space, our model attains a superior position, especially with its transformer-based architecture.

## 5 CONCLUSION

We introduced EntroPE, a temporally informed dynamic patch encoding framework that extends time series transformer architectures through entropy-guided boundary detection. Instead of partitioning sequences at arbitrary intervals, EntroPE strategically places patch boundaries at natural temporal transitions, preserving temporal coherence while supporting efficient variable-length patch processing. This design provides a principled way to incorporate information-theoretic signals into patching, yielding improved forecasting accuracy alongside practical computational benefits. Beyond its empirical performance, EntroPE highlights the importance of respecting the intrinsic structure of time series, opening new research directions in adaptive sequence modeling and context-aware temporal representation learning.

REPRODUCIBILITY STATEMENT

All experimental results reported in this paper can be fully reproduced using the code provided at `https://anonymous.4open.science/r/EntroPE-501C/`. The implementation includes complete details of model training, data preprocessing, and evaluation pipelines. All hyperparameter configurations are documented both in the released codebase and in Appendix A.4, ensuring transparent replication of our findings. Links to the original benchmark datasets along with instructions for downloading are included in the repository README file.

All experiments are conducted with 5 different random seeds, and reported results correspond to the best performance metrics across these runs. Experiments were performed on multiple GPU platforms, including NVIDIA RTX 4090, RTX A5000, NVIDIA GeForce RTX 4090 D, and NVIDIA RTX 6000 Ada-16Q. Despite hardware differences, the experimental pipeline is designed to remain robust and reproducible, with expected training and evaluation runtimes reported in the repository.

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

# A    APPENDIX

## A.1    USE OF LARGE LANGUAGE MODELS

Parts of the text in this paper were refined with the assistance of a large language model (ChatGPT, GPT-5), used exclusively for language polishing and improving clarity of exposition. All ideas, methodology, experiments, and analyses were conceived and executed solely by the authors.

## A.2    BASELINE DETAILS

Table 3: Overview of baseline models (2021–2025) grouped by architecture. We highlight their key design paradigms for deep time series forecasting.

| Model | Category | Key Features / Innovations |
|---|---|---|
| iTransformer (Liu et al., 2024b) | Transformer | Inverted attention across feature and time dimensions. |
| PatchTST (Nie et al., 2023) | Transformer | Patch-based embedding with channel-independence. |
| FEDformer (Zhou et al., 2022) | Transformer (freq.-decomp.) | Fourier/Wavelet enhanced decomposition. |
| Autoformer (Wu et al., 2021) | Transformer (decomp.-autocorr.) | Progressive decomposition with auto-correlation. |
| TimesNet (Wu et al., 2023) | CNN | Temporal 2D variation modeling with convolutional layers. |
| TimeMixer (Wang et al., 2024) | MLP | Multiscale mixing, separation of trend and seasonal signals. |
| HDMixer (Huang et al., 2024) | MLP | Hierarchical dependency with extendable patches. |
| DLinear (Zeng et al., 2023) | Linear/MLP | Channel-independent trend/seasonal decomposition. |
| TimeKAN (Huang et al., 2025b) | Kernel Attn. Network | Kernelized attention distinct from Transformer/MLP. |
| Time-FFM (Liu et al., 2024a) | Foundation | Pre-trained foundation model for time series. |
| TimeBase (Huang et al., 2025a) | Lightweight | Resource-efficient, practical forecasting baseline. |
| LangTime (Niu et al., 2025) | LLM-aligned / Cross-modal | Language-guided forecasting with RL optimization. |
| CALF (Liu et al., 2025) | LLM-aligned / Cross-modal | Cross-modal match, feature regularization, output consistency. |
| FilterTS (Wang et al., 2025) | Filter-based | Static global + dynamic cross-variable frequency filtering. |

## A.3    DATASET DETAILS

This study employs a comprehensive collection of benchmark forecasting datasets to rigorously evaluate model performance across diverse temporal domains and application scenarios. The selected datasets represent critical real-world forecasting challenges spanning energy management, meteorological prediction, and financial markets.

- **Electricity Transformer Temperature (ETT) Datasets:** The ETT collection comprises four distinct datasets (ETTh1, ETTh2, ETTm1, and ETTm2) that monitor temperature variations in electricity transformers alongside corresponding load measurements. These datasets facilitate the prediction of future temperature profiles and electrical loads based on historical patterns, which is essential for transformer maintenance and grid stability. The collection offers varied temporal granularities: ETTh1 and ETTh2 provide hourly measurements, while ETTm1 and ETTm2 capture data at 15-minute intervals, enabling comprehensive evaluation across different forecasting horizons and temporal resolutions.

- **Weather:** The weather dataset incorporates comprehensive meteorological measurements recorded at 10-minute intervals from a dedicated weather station. This dataset supports forecasting of various atmospheric phenomena, providing crucial information for agricultural planning, transportation safety, and general societal planning activities. The multivariate nature of weather data makes it particularly challenging for forecasting models, as it requires capturing complex interdependencies among atmospheric variables.

- **Electricity (ECL):** This dataset encompasses hourly electricity consumption records from 321 individual clients, providing comprehensive insights into consumption patterns and enabling accurate demand forecasting. The dataset is particularly valuable for optimizing power generation scheduling and distribution network management, representing a critical application domain for time series forecasting in energy systems.

- **Exchange Rate:** This dataset contains daily exchange rates of the US dollar against eight different currencies, spanning from 1990 to 2016. It represents financial time series forecasting challenges with inherent volatility and complex market dynamics.

Table 4 presents the detailed characteristics of all datasets employed in this study. The collection spans multiple temporal frequencies (10-minute, 15-minute, hourly, and daily intervals) and varies significantly in dimensionality, from 7-dimensional ETT datasets to the 321-dimensional ECL dataset. Dataset sizes are reported in the format (Training, Validation, Test) to clearly delineate the data partitioning strategy employed across all experiments.

Table 4: Summary of datasets used for long-term forecasting evaluation. Dataset sizes represent the number of temporal observations in each partition (Training, Validation, Test).

| Dataset | Dim | Dataset Size (Train, Val, Test) | Frequency | Domain Information |
|---|---|---|---|---|
| ETTh1 | 7 | (8545, 2881, 2881) | Hourly | Energy Infrastructure |
| ETTh2 | 7 | (8545, 2881, 2881) | Hourly | Energy Infrastructure |
| ETTm1 | 7 | (34465, 11521, 11521) | 15min | Energy Infrastructure |
| ETTm2 | 7 | (34465, 11521, 11521) | 15min | Energy Infrastructure |
| Weather | 21 | (36792, 5271, 10540) | 10min | Meteorological Monitoring |
| Electricity (ECL) | 321 | (18317, 2633, 5261) | Hourly | Electricity Consumption |
| Exchange Rate | 8 | (5120, 665, 1422) | Daily | Foreign Exchange Market |

## A.4 IMPLEMENTATION DETAILS

We summarize the hyperparameter configurations used for EntroPE across all benchmark datasets in Table 6. The table reports dataset-specific settings alongside common defaults, covering patching, embedding, optimization, and training parameters to ensure reproducibility and fair comparison with prior work.

All experiments are conducted with 5 different random seeds, and reported results correspond to the best performance metrics across these runs. Experiments were performed on multiple GPU platforms, including NVIDIA RTX 4090, RTX A5000, NVIDIA GeForce RTX 4090 D, and NVIDIA RTX 6000 Ada-16Q.

**Entropy Model Pre-Training.** We adopt the exact configuration described in the EDP section 3.3. The model hyperparameters are summarized in Table 5. We pre-train this lightweight model with early stopping, where training is terminated once the validation loss fails to improve by more than 7%. The same setting is applied across all datasets, except for the Electricity dataset where we use a batch size of 32 (batch size of 128 is used for all others).

Table 5: Entropy model configuration used for pre-training.

| Hyperparameter | Value |
|---|---|
| Number of layers ($n\_layer$) | 2 |
| Number of heads ($n\_head$) | 4 |
| Embedding dimension ($n\_embd$) | 8 |
| Dropout | 0.1 |
| Bias | False |
| Vocabulary size ($vocab\_size$) | 256 |
| Block size ($block\_size$) | 96 |

## A.5 EXTENDED RESULTS

### A.5.1 FULL RESULTS (LONG-TERM FORECASTING). INPUT LENGTH 96.

We adopted the training criteria established by TimeKAN for fair comparison across all baseline methods. The results for AutoFormer, FEDformer, TimesNet, PatchTST, Time-FFM, iTransformer, and TimeMixer were directly extracted from the TimeKAN paper to ensure consistency in experimental conditions. For HDMixer, TimeBase, CALF, and FilterTS, we executed the original implementations following identical training criteria. Specifically for FilterTS, we addressed the challenge

Table 6: Hyperparameter settings for EntroPE across datasets.

| Hyperparameter | ETTh1 | ETTh2 | ETTm1 | ETTm2 | Weather | Electricity | Exchange Rate |
|---|---|---|---|---|---|---|---|
| dim | 8 | 8 | 16 | 16 | 16 | 32 | 8 |
| heads | 2 | 2 | 2 | 4 | 2 | 4 | 2 |
| layers | 1 | 2 | 1 | 1 | 2 | 2 | 1 |
| max patch length | 24 | 24 | 24 | 24 | 24 | 24 | 16 |
| batch_size | 64 | 64 | 32 | 32 | 128 | 32 | 32 |
| learning_rate | 0.001 | 0.01 | 0.01 | 0.001 | 0.01 | 0.01 | 0.01 |
| dropout | 0.05 | 0.1 | 0.1 | 0.1 | 0.2 | 0.1 | 0.2 |
| monotonicity | 1 | 1 | 1 | 1 | 1 | 1 | 1 |
| threshold ($\theta$) | 3 | 3.5 | 3.5 | 3 | 3 | 3.5 | 3.5 |
| threshold_add ($\gamma$) | 0.25 | 0.2 | 0.3 | 0.25 | 0.35 | 0.3 | 0.3 |
| train_epochs | 20 | 20 | 20 | 20 | 20 | 20 | 20 |
| patience | 7 | 7 | 7 | 7 | 7 | 7 | 7 |

of multiple configuration options by running experiments across all different settings proposed by the authors for each forecasting horizon. We then selected the unified setting that achieved the lowest Mean Squared Error (MSE) and Mean Absolute Error (MAE) performance (detailed results are provided in Section A.5.2). LangTime baseline results were obtained directly by running the author's implementation to maintain consistency with the their reported performance. This methodology ensures that all comparative results are obtained under equivalent experimental conditions, providing a fair and rigorous evaluation framework for our proposed approach. See Tab 7 and Tab. 8 for full results.

### A.5.2    FILTERTS PERFORMANCE COMPARISON ACROSS CONFIGURATIONS

The original FilterTS results (9) were reported using different hyperparameter settings for different forecast lengths, which complicates direct comparison with other baselines. To ensure fairness, we systematically re-evaluate all provided configurations (Config 1-4) across all forecast horizons and report the configuration that achieves the best performance in terms of MSE and MAE.

### A.6    ENTROPY QUALITATIVE ANALYSIS

**Entropy Visualization:** Figure 9 demonstrates how entropy-driven boundaries align with temporal transitions, creating patches that preserve intra-patch coherence while capturing natural temporal structure. This validates our core hypothesis that content-aware segmentation outperforms arbitrary fixed-length approaches.

In figure 9,
**Panel (a) - Original Time Series:** Shows the normalized time series data from the ETTh1 dataset (Sample 0), exhibiting typical temporal patterns with periodic fluctuations and trend changes. The series demonstrates varying levels of predictability across different time segments, ranging from smooth transitions to sharp discontinuities.
**Panel (b) - Tokenized Sequence:** Displays the discrete token representation of the continuous time series after quantization using the MeanScaleUniformBins tokenizer. Token values range approximately from 50 to 180, capturing the underlying temporal dynamics while enabling discrete sequence modeling. The step-wise pattern reflects the quantization process that maps continuous values to discrete vocabulary tokens.
**Panel (c) - Token-wise Entropy with Threshold Regions:** Presents the entropy values computed by the pre-trained entropy model for each token position. High entropy regions (red shading) indi-

Table 7: Forecasting performance comparison (MSE and MAE) across baselines and our proposed EntroPE model. Look-back window $L$ set to 96 for all cases, forecast window $T = \{96, 192, 336, 720\}$

| Dataset | T | EntroPE | | TimeKAN | | HDMixer | | TimeBase | | CALF | | FilterTS | | LangTime | | TimeMixer | | Time-FFM | |
|---|---|---|---|---|---|---|---|---|---|---|---|---|---|---|---|---|---|---|---|
| | | MSE | MAE | MSE | MAE | MSE | MAE | MSE | MAE | MSE | MAE | MSE | MAE | MSE | MAE | MSE | MAE | MSE | MAE |
| ETTm1 | 96 | 0.317 | 0.354 | 0.327 | 0.365 | 0.340 | 0.369 | 0.373 | 0.388 | 0.319 | 0.348 | 0.323 | 0.362 | 0.319 | 0.348 | 0.317 | 0.356 | 0.336 | 0.369 |
| | 192 | 0.360 | 0.379 | 0.354 | 0.380 | 0.379 | 0.385 | 0.411 | 0.409 | 0.375 | 0.376 | 0.364 | 0.382 | 0.368 | 0.375 | 0.367 | 0.384 | 0.378 | 0.389 |
| | 336 | 0.388 | 0.399 | 0.386 | 0.403 | 0.398 | 0.409 | 0.436 | 0.421 | 0.410 | 0.399 | 0.396 | 0.404 | 0.413 | 0.402 | 0.391 | 0.406 | 0.411 | 0.410 |
| | 720 | 0.446 | 0.433 | 0.452 | 0.442 | 0.467 | 0.443 | 0.503 | 0.461 | 0.478 | 0.439 | 0.462 | 0.441 | 0.487 | 0.439 | 0.454 | 0.441 | 0.469 | 0.441 |
| | **Avg.** | **0.378** | **0.391** | 0.380 | 0.398 | 0.396 | 0.402 | 0.431 | 0.420 | 0.396 | 0.391 | 0.386 | 0.397 | 0.397 | 0.392 | 0.382 | 0.397 | 0.399 | 0.402 |
| ETTm2 | 96 | 0.180 | 0.265 | 0.175 | 0.258 | 0.183 | 0.266 | 0.188 | 0.271 | 0.174 | 0.252 | 0.174 | 0.256 | 0.188 | 0.258 | 0.175 | 0.257 | 0.181 | 0.267 |
| | 192 | 0.241 | 0.311 | 0.241 | 0.301 | 0.246 | 0.306 | 0.251 | 0.309 | 0.241 | 0.297 | 0.240 | 0.299 | 0.245 | 0.297 | 0.240 | 0.302 | 0.247 | 0.308 |
| | 336 | 0.313 | 0.352 | 0.308 | 0.349 | 0.307 | 0.347 | 0.311 | 0.346 | 0.304 | 0.338 | 0.301 | 0.339 | 0.301 | 0.336 | 0.303 | 0.343 | 0.309 | 0.347 |
| | 720 | 0.411 | 0.413 | 0.417 | 0.414 | 0.408 | 0.404 | 0.411 | 0.401 | 0.402 | 0.395 | 0.399 | 0.399 | 0.402 | 0.393 | 0.392 | 0.396 | 0.406 | 0.404 |
| | **Avg.** | 0.286 | 0.335 | 0.285 | 0.331 | 0.286 | 0.331 | 0.290 | 0.332 | **0.280** | **0.321** | 0.279 | 0.323 | 0.284 | 0.321 | 0.279 | 0.324 | 0.286 | 0.332 |
| ETTh1 | 96 | 0.375 | 0.399 | 0.367 | 0.395 | 0.387 | 0.401 | 0.399 | 0.392 | 0.377 | 0.395 | 0.377 | 0.391 | 0.391 | 0.388 | 0.385 | 0.402 | 0.385 | 0.400 |
| | 192 | 0.423 | 0.425 | 0.416 | 0.424 | 0.441 | 0.428 | 0.455 | 0.423 | 0.429 | 0.427 | 0.431 | 0.423 | 0.429 | 0.419 | 0.443 | 0.430 | 0.439 | 0.430 |
| | 336 | 0.429 | 0.432 | 0.446 | 0.436 | 0.452 | 0.433 | 0.501 | 0.443 | 0.475 | 0.449 | 0.479 | 0.448 | 0.462 | 0.440 | 0.512 | 0.470 | 0.480 | 0.449 |
| | 720 | 0.439 | 0.454 | 0.441 | 0.453 | 0.513 | 0.485 | 0.498 | 0.458 | 0.482 | 0.470 | 0.471 | 0.466 | 0.458 | 0.445 | 0.497 | 0.496 | 0.462 | 0.456 |
| | **Avg.** | **0.416** | **0.425** | 0.418 | 0.427 | 0.448 | 0.437 | 0.463 | 0.429 | 0.441 | 0.435 | 0.440 | 0.432 | 0.437 | 0.425 | 0.459 | 0.444 | 0.442 | 0.434 |
| ETTh2 | 96 | 0.281 | 0.336 | 0.290 | 0.339 | 0.289 | 0.336 | 0.338 | 0.376 | 0.288 | 0.336 | 0.293 | 0.344 | 0.299 | 0.336 | 0.289 | 0.342 | 0.301 | 0.351 |
| | 192 | 0.371 | 0.393 | 0.386 | 0.399 | 0.386 | 0.397 | 0.402 | 0.405 | 0.368 | 0.386 | 0.374 | 0.389 | 0.374 | 0.382 | 0.378 | 0.397 | 0.378 | 0.397 |
| | 336 | 0.392 | 0.394 | 0.441 | 0.447 | 0.438 | 0.442 | 0.437 | 0.440 | 0.415 | 0.423 | 0.411 | 0.423 | 0.410 | 0.418 | 0.432 | 0.434 | 0.422 | 0.431 |
| | 720 | 0.421 | 0.427 | 0.450 | 0.458 | 0.421 | 0.454 | 0.460 | 0.477 | 0.417 | 0.435 | 0.423 | 0.441 | 0.418 | 0.426 | 0.464 | 0.464 | 0.427 | 0.444 |
| | **Avg.** | **0.366** | **0.387** | 0.391 | 0.410 | 0.384 | 0.407 | 0.409 | 0.425 | 0.372 | 0.395 | 0.375 | 0.399 | 0.375 | 0.392 | 0.390 | 0.409 | 0.382 | 0.406 |
| Weather | 96 | 0.164 | 0.211 | 0.163 | 0.209 | 0.175 | 0.223 | 0.170 | 0.215 | 0.164 | 0.205 | 0.161 | 0.208 | 0.178 | 0.204 | 0.163 | 0.209 | 0.191 | 0.230 |
| | 192 | 0.210 | 0.252 | 0.208 | 0.251 | 0.226 | 0.265 | 0.216 | 0.256 | 0.213 | 0.252 | 0.226 | 0.264 | 0.211 | 0.249 | 0.211 | 0.254 | 0.236 | 0.267 |
| | 336 | 0.256 | 0.290 | 0.263 | 0.290 | 0.264 | 0.301 | 0.272 | 0.297 | 0.272 | 0.274 | 0.279 | 0.304 | 0.269 | 0.292 | 0.263 | 0.293 | 0.289 | 0.303 |
| | 720 | 0.339 | 0.342 | 0.341 | 0.341 | 0.348 | 0.349 | 0.351 | 0.348 | 0.352 | 0.346 | 0.345 | 0.344 | 0.351 | 0.347 | 0.344 | 0.348 | 0.362 | 0.350 |
| | **Avg.** | **0.242** | **0.273** | 0.244 | 0.273 | 0.253 | 0.285 | 0.252 | 0.279 | 0.250 | 0.274 | 0.253 | 0.280 | 0.252 | 0.273 | 0.245 | 0.276 | 0.270 | 0.288 |
| Electricity | 96 | 0.163 | 0.252 | 0.174 | 0.266 | 0.180 | 0.271 | 0.212 | 0.279 | 0.145 | 0.239 | 0.153 | 0.247 | 0.181 | 0.266 | 0.153 | 0.245 | 0.198 | 0.282 |
| | 192 | 0.177 | 0.268 | 0.182 | 0.273 | 0.184 | 0.275 | 0.209 | 0.281 | 0.162 | 0.255 | 0.168 | 0.260 | 0.185 | 0.273 | 0.166 | 0.257 | 0.199 | 0.285 |
| | 336 | 0.194 | 0.284 | 0.197 | 0.286 | 0.207 | 0.299 | 0.222 | 0.295 | 0.177 | 0.268 | 0.187 | 0.278 | 0.198 | 0.281 | 0.185 | 0.275 | 0.212 | 0.298 |
| | 720 | 0.235 | 0.321 | 0.236 | 0.320 | 0.249 | 0.335 | 0.264 | 0.327 | 0.222 | 0.304 | 0.227 | 0.313 | 0.241 | 0.320 | 0.224 | 0.312 | 0.253 | 0.330 |
| | **Avg.** | 0.182 | 0.271 | 0.197 | 0.286 | 0.205 | 0.295 | 0.227 | 0.296 | **0.177** | **0.266** | 0.184 | 0.275 | 0.201 | 0.285 | 0.182 | 0.272 | 0.216 | 0.299 |
| Exchange | 96 | 0.083 | 0.140 | – | – | – | – | – | – | – | – | 0.084 | 0.201 | 0.089 | 0.201 | 0.087 | 0.205 | 0.081 | 0.201 |
| | 192 | 0.177 | 0.235 | – | – | – | – | – | – | – | – | 0.173 | 0.295 | 0.175 | 0.298 | 0.179 | 0.300 | 0.168 | 0.293 |
| | 336 | 0.299 | 0.371 | – | – | – | – | – | – | – | – | 0.316 | 0.407 | 0.329 | 0.409 | 0.333 | 0.417 | 0.299 | 0.396 |
| | 720 | 0.761 | 0.869 | – | – | – | – | – | – | – | – | 0.827 | 0.685 | 0.852 | 0.690 | 0.912 | 0.719 | 0.805 | 0.674 |
| | **Avg.** | **0.331** | **0.386** | – | – | – | – | – | – | – | – | 0.355 | 0.400 | 0.361 | 0.400 | 0.378 | 0.410 | 0.338 | 0.391 |

cate positions where the next token is highly uncertain, suggesting natural breakpoints for dynamic patching. Low entropy regions (green shading) represent predictable sequences that can be grouped into coherent patches. The entropy fluctuates between approximately 3.2-4.4 nats, with clear temporal patterns corresponding to the underlying time series structure.

**Panel (d) - Time Series with Entropy-based Coloring:** Overlays the original time series with color-coding based on entropy values. Yellow points indicate high entropy (high uncertainty), while dark blue/brown points represent low entropy (high predictability). This visualization reveals the relationship between time series patterns and predictive uncertainty, where rapid transitions and trend changes typically correspond to higher entropy values.

## A.7 TIME SERIES QUANTIZATION

Real-valued time series data cannot be directly processed by entropy-based boundary detection methods that rely on discrete probability distributions. Following Ansari et al. (2024), we employ quantization to convert continuous values into discrete tokens while preserving temporal structure.

Given a scaled time series $\tilde{x}_{1:C+H} = [\tilde{x}_1, \ldots, \tilde{x}_C, \ldots, \tilde{x}_{C+H}]$, we define a quantization function $q : \mathbb{R} \to \{1, 2, \ldots, B\}$ that maps real values to discrete bins. The quantization and dequantization functions are defined as:

$$q(x) = \begin{cases} 1 & \text{if } -\infty \leq x < b_1, \\ 2 & \text{if } b_1 \leq x < b_2, \\ \vdots & \\ B & \text{if } b_{B-1} \leq x < \infty, \end{cases} \quad \text{and} \quad d(j) = c_j \tag{10}$$

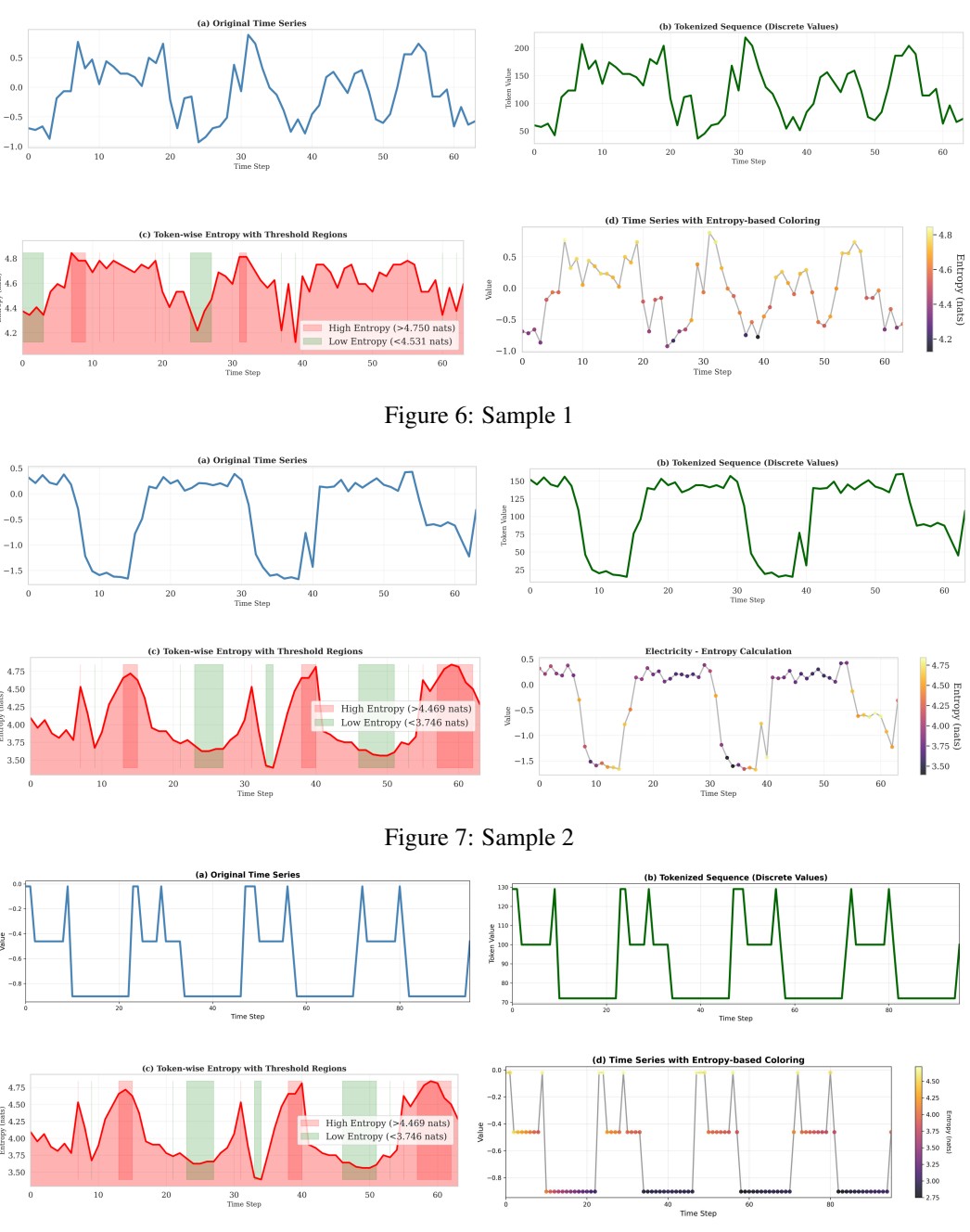

Figure 9: Comprehensive entropy analysis for dynamic patching on ETT and Electricity data. Sub-figures show how entropy-driven patches align with temporal transitions.

Table 8: Forecasting performance comparison (MSE and MAE) across baselines and our proposed **EntroPE** model. The look-back window is fixed at $L = 96$, with forecast horizons $T = \{96, 192, 336, 720\}$. Baseline results up to 2024 are extracted from Huang et al. (2025b).

| Dataset | T | **EntroPE** | | iTrans. | | PatchTST | | TimesNet | | DLinear | | FEDformer | | Autoformer | |
|---|---|---|---|---|---|---|---|---|---|---|---|---|---|---|---|
| | | MSE | MAE | MSE | MAE | MSE | MAE | MSE | MAE | MSE | MAE | MSE | MAE | MSE | MAE |
| ETTm1 | 96 | 0.317 | 0.354 | 0.334 | 0.368 | 0.352 | 0.374 | 0.338 | 0.375 | 0.346 | 0.374 | 0.379 | 0.419 | 0.505 | 0.475 |
| | 192 | 0.360 | 0.379 | 0.377 | 0.391 | 0.390 | 0.393 | 0.374 | 0.387 | 0.382 | 0.391 | 0.426 | 0.441 | 0.553 | 0.496 |
| | 336 | 0.388 | 0.399 | 0.426 | 0.420 | 0.421 | 0.414 | 0.410 | 0.411 | 0.415 | 0.415 | 0.445 | 0.459 | 0.621 | 0.537 |
| | 720 | 0.446 | 0.433 | 0.491 | 0.459 | 0.462 | 0.449 | 0.478 | 0.450 | 0.473 | 0.451 | 0.543 | 0.490 | 0.671 | 0.561 |
| | Avg. | **0.378** | **0.391** | 0.407 | 0.410 | 0.406 | 0.407 | 0.400 | 0.406 | 0.404 | 0.408 | 0.448 | 0.452 | 0.588 | 0.517 |
| ETTm2 | 96 | 0.180 | 0.265 | 0.180 | 0.264 | 0.183 | 0.270 | 0.187 | 0.267 | 0.193 | 0.293 | 0.203 | 0.287 | 0.255 | 0.339 |
| | 192 | 0.241 | 0.311 | 0.250 | 0.309 | 0.255 | 0.314 | 0.249 | 0.309 | 0.284 | 0.361 | 0.269 | 0.328 | 0.281 | 0.340 |
| | 336 | 0.313 | 0.352 | 0.311 | 0.348 | 0.309 | 0.347 | 0.321 | 0.351 | 0.382 | 0.429 | 0.325 | 0.366 | 0.339 | 0.372 |
| | 720 | 0.411 | 0.413 | 0.412 | 0.407 | 0.412 | 0.404 | 0.408 | 0.403 | 0.558 | 0.525 | 0.421 | 0.415 | 0.433 | 0.432 |
| | Avg. | **0.286** | 0.335 | 0.288 | **0.332** | 0.290 | 0.334 | 0.291 | 0.333 | 0.354 | 0.402 | 0.305 | 0.349 | 0.327 | 0.371 |
| ETTh1 | 96 | 0.375 | 0.399 | 0.386 | 0.405 | 0.460 | 0.447 | 0.384 | 0.402 | 0.397 | 0.412 | 0.395 | 0.424 | 0.449 | 0.459 |
| | 192 | 0.423 | 0.425 | 0.441 | 0.436 | 0.512 | 0.477 | 0.436 | 0.429 | 0.446 | 0.441 | 0.469 | 0.470 | 0.500 | 0.482 |
| | 336 | 0.429 | 0.432 | 0.487 | 0.458 | 0.546 | 0.496 | 0.638 | 0.469 | 0.489 | 0.467 | 0.490 | 0.477 | 0.521 | 0.496 |
| | 720 | 0.439 | 0.454 | 0.503 | 0.491 | 0.544 | 0.517 | 0.521 | 0.500 | 0.513 | 0.510 | 0.598 | 0.544 | 0.514 | 0.512 |
| | Avg. | **0.416** | **0.425** | 0.454 | 0.447 | 0.516 | 0.484 | 0.495 | 0.450 | 0.461 | 0.457 | 0.498 | 0.484 | 0.496 | 0.487 |
| ETTh2 | 96 | 0.281 | 0.336 | 0.297 | 0.349 | 0.308 | 0.355 | 0.340 | 0.374 | 0.340 | 0.394 | 0.358 | 0.397 | 0.346 | 0.388 |
| | 192 | 0.371 | 0.393 | 0.380 | 0.400 | 0.393 | 0.405 | 0.402 | 0.414 | 0.482 | 0.479 | 0.429 | 0.439 | 0.456 | 0.452 |
| | 336 | 0.392 | 0.394 | 0.428 | 0.432 | 0.427 | 0.436 | 0.452 | 0.452 | 0.591 | 0.541 | 0.496 | 0.487 | 0.482 | 0.486 |
| | 720 | 0.421 | 0.427 | 0.427 | 0.445 | 0.436 | 0.450 | 0.462 | 0.468 | 0.839 | 0.661 | 0.463 | 0.474 | 0.515 | 0.511 |
| | Avg. | **0.366** | **0.387** | 0.383 | 0.407 | 0.391 | 0.411 | 0.414 | 0.427 | 0.563 | 0.519 | 0.437 | 0.449 | 0.450 | 0.459 |
| Weather | 96 | 0.164 | 0.211 | 0.174 | 0.214 | 0.186 | 0.227 | 0.172 | 0.220 | 0.195 | 0.252 | 0.217 | 0.296 | 0.266 | 0.336 |
| | 192 | 0.210 | 0.252 | 0.221 | 0.254 | 0.234 | 0.265 | 0.219 | 0.261 | 0.237 | 0.295 | 0.276 | 0.336 | 0.307 | 0.367 |
| | 336 | 0.256 | 0.290 | 0.278 | 0.296 | 0.284 | 0.301 | 0.246 | 0.337 | 0.282 | 0.331 | 0.339 | 0.380 | 0.359 | 0.395 |
| | 720 | 0.339 | 0.342 | 0.358 | 0.347 | 0.356 | 0.349 | 0.365 | 0.359 | 0.345 | 0.382 | 0.403 | 0.428 | 0.419 | 0.428 |
| | Avg. | **0.242** | **0.273** | 0.258 | 0.278 | 0.265 | 0.285 | 0.251 | 0.294 | 0.265 | 0.315 | 0.309 | 0.360 | 0.338 | 0.382 |
| Electricity | 96 | 0.163 | 0.252 | 0.148 | 0.240 | 0.190 | 0.296 | 0.168 | 0.272 | 0.210 | 0.302 | 0.193 | 0.308 | 0.201 | 0.317 |
| | 192 | 0.177 | 0.268 | 0.162 | 0.253 | 0.199 | 0.304 | 0.184 | 0.322 | 0.210 | 0.305 | 0.201 | 0.315 | 0.222 | 0.334 |
| | 336 | 0.194 | 0.284 | 0.178 | 0.269 | 0.217 | 0.319 | 0.198 | 0.300 | 0.223 | 0.319 | 0.214 | 0.329 | 0.231 | 0.443 |
| | 720 | 0.235 | 0.321 | 0.225 | 0.317 | 0.258 | 0.352 | 0.220 | 0.320 | 0.258 | 0.350 | 0.246 | 0.355 | 0.254 | 0.361 |
| | Avg. | 0.182 | 0.271 | **0.178** | **0.270** | 0.216 | 0.318 | 0.193 | 0.304 | 0.225 | 0.319 | 0.214 | 0.327 | 0.227 | 0.338 |
| Exchange | 96 | 0.083 | 0.140 | 0.086 | 0.206 | 0.088 | 0.205 | 0.107 | 0.234 | 0.088 | 0.218 | 0.148 | 0.278 | 0.197 | 0.323 |
| | 192 | 0.177 | 0.235 | 0.177 | 0.299 | 0.176 | 0.299 | 0.226 | 0.344 | 0.176 | 0.315 | 0.271 | 0.315 | 0.300 | 0.369 |
| | 336 | 0.299 | 0.371 | 0.331 | 0.417 | 0.301 | 0.397 | 0.367 | 0.448 | 0.313 | 0.427 | 0.460 | 0.427 | 0.509 | 0.524 |
| | 720 | 0.761 | 0.869 | 0.847 | 0.691 | 0.901 | 0.714 | 0.964 | 0.746 | 0.839 | 0.695 | 1.195 | 0.695 | 1.447 | 0.941 |
| | Avg. | **0.331** | **0.386** | 0.360 | 0.403 | 0.367 | 0.404 | 0.416 | 0.443 | 0.354 | 0.414 | 0.519 | 0.429 | 0.613 | 0.539 |

Unlike traditional approaches that rely on dequantization for prediction, our method uses quantization purely for entropy-based boundary detection while employing linear projection on continuous representations for forecasting.

## A.8 ENTROPY CALCULATION

We begin with the standard Shannon entropy for a discrete random variable $X$ with possible values $\{x_1, x_2, \ldots, x_n\}$:

$$H(X) = -\sum_{i=1}^{n} p(x_i) \log p(x_i) \tag{11}$$

For sequential data, we are interested in the entropy of a variable $X_{i+1}$ conditioned on the previous observations $X_{\leq i} = \{X_1, X_2, \ldots, X_i\}$. This leads to the conditional entropy:

$$H(X_{i+1}|X_{\leq i}) = -\sum_{v \in \mathcal{V}} p(X_{i+1} = v|X_{\leq i}) \log p(X_{i+1} = v|X_{\leq i}) \tag{12}$$

Table 9: Comprehensive evaluation of FilterTS across multiple hyperparameter configurations. We assess all available configurations (Config 1-4) for each forecast horizon and report the best-performing results based on MSE and MAE. Missing values (-) denote configurations not originally reported for certain horizons in the official implementation.

| Dataset | L | T | Config 1 | | Config 2 | | Config 3 | | Config 4 | |
|---|---|---|---|---|---|---|---|---|---|---|
| | | | MSE | MAE | MSE | MAE | MSE | MAE | MSE | MAE |
| ETTm1 | 96 | 96 | 0.321 | 0.360 | 0.323 | 0.362 | 0.330 | 0.366 | – | – |
| | 96 | 192 | 0.363 | 0.382 | 0.364 | 0.382 | 0.366 | 0.383 | – | – |
| | 96 | 336 | 0.397 | 0.405 | 0.396 | 0.404 | 0.398 | 0.403 | – | – |
| | 96 | 720 | 0.478 | 0.447 | 0.462 | 0.441 | 0.462 | 0.438 | – | – |
| | | Avg. | 0.390 | 0.399 | 0.386 | 0.397 | 0.389 | 0.398 | – | – |
| ETTm2 | 96 | 96 | 0.174 | 0.256 | 0.174 | 0.257 | 0.174 | 0.256 | – | – |
| | 96 | 192 | 0.239 | 0.300 | 0.238 | 0.299 | 0.240 | 0.299 | – | – |
| | 96 | 336 | 0.299 | 0.338 | 0.300 | 0.340 | 0.301 | 0.339 | – | – |
| | 96 | 720 | 0.405 | 0.399 | 0.406 | 0.399 | 0.399 | 0.399 | – | – |
| | | Avg. | 0.279 | 0.323 | 0.280 | 0.324 | 0.279 | 0.323 | – | – |
| ETTh1 | 96 | 96 | 0.375 | 0.390 | 0.381 | 0.396 | 0.377 | 0.391 | – | – |
| | 96 | 192 | 0.424 | 0.421 | 0.431 | 0.423 | 0.431 | 0.423 | – | – |
| | 96 | 336 | 0.479 | 0.451 | 0.465 | 0.442 | 0.479 | 0.448 | – | – |
| | 96 | 720 | 0.480 | 0.471 | 0.495 | 0.481 | 0.471 | 0.466 | – | – |
| | | Avg. | 0.440 | 0.433 | 0.443 | 0.436 | 0.440 | 0.432 | – | – |
| ETTh2 | 96 | 96 | 0.289 | 0.338 | 0.293 | 0.344 | 0.291 | 0.341 | – | – |
| | 96 | 192 | 0.374 | 0.390 | 0.374 | 0.389 | 0.394 | 0.400 | – | – |
| | 96 | 336 | 0.416 | 0.426 | 0.411 | 0.423 | 0.433 | 0.431 | – | – |
| | 96 | 720 | 0.421 | 0.440 | 0.423 | 0.441 | 0.431 | 0.442 | – | – |
| | | Avg. | 0.375 | 0.399 | 0.375 | 0.399 | 0.387 | 0.404 | – | – |
| Weather | 96 | 96 | 0.161 | 0.208 | 0.180 | 0.228 | 0.182 | 0.228 | 0.161 | 0.208 |
| | 96 | 192 | 0.226 | 0.264 | 0.210 | 0.252 | 0.213 | 0.255 | 0.226 | 0.264 |
| | 96 | 336 | 0.279 | 0.304 | 0.286 | 0.307 | 0.263 | 0.292 | 0.279 | 0.304 |
| | 96 | 720 | 0.345 | 0.344 | 0.349 | 0.347 | 0.351 | 0.348 | 0.345 | 0.344 |
| | | Avg. | 0.253 | 0.280 | 0.256 | 0.284 | 0.252 | 0.281 | 0.253 | 0.280 |
| Electricity | 96 | 96 | 0.151 | 0.245 | 0.153 | 0.247 | – | – | – | – |
| | 96 | 192 | 0.164 | 0.256 | 0.168 | 0.260 | – | – | – | – |
| | 96 | 336 | 0.181 | 0.274 | 0.187 | 0.278 | – | – | – | – |
| | 96 | 720 | 0.241 | 0.326 | 0.227 | 0.313 | – | – | – | – |
| | | Avg. | 0.184 | 0.275 | 0.184 | 0.275 | – | – | – | – |
| Exchange Rate | 96 | 96 | 0.085 | 0.201 | 0.083 | 0.200 | 0.084 | 0.201 | 0.082 | 0.198 |
| | 96 | 192 | 0.180 | 0.300 | 0.171 | 0.294 | 0.173 | 0.295 | 0.172 | 0.294 |
| | 96 | 336 | 0.342 | 0.420 | 0.323 | 0.411 | 0.316 | 0.407 | 0.322 | 0.410 |
| | 96 | 720 | 0.878 | 0.699 | 0.916 | 0.713 | 0.827 | 0.685 | 0.828 | 0.686 |
| | | Avg. | 0.371 | 0.405 | 0.373 | 0.405 | 0.350 | 0.397 | 0.351 | 0.397 |

In practice, we estimate these conditional probabilities using a parameterized model $\theta$, giving us:

$$H(x_i) = -\sum_{v \in \mathcal{V}} p_\theta(x_{i+1} = v | x_{\leq i}) \log p_\theta(x_{i+1} = v | x_{\leq i}) \tag{13}$$

where:

- $H(x_i)$ represents the conditional entropy at position $i$

- $p_\theta(x_{i+1} = v | x_{\leq i})$ is the model's predicted probability distribution over the vocabulary $\mathcal{V}$

- $x_{\leq i} = \{x_1, x_2, \ldots, x_i\}$ denotes all observations up to position $i$

- $\theta$ represents the learnable parameters of the predictive model

This conditional entropy quantifies the uncertainty in predicting the next value given the historical context, with higher values indicating less predictable (more informative) regions of the time series.

A.9   CROSS-ENTROPY LOSS

$$L_{CE} = -\sum_{i=1}^{N-1} \log P(t_{i+1}|t_1, \ldots, t_i; \theta) \tag{14}$$

where $\theta$ represents the model parameters.

A.10   MEAN SQUARED ERROR (MSE)

$$L_{MSE} = \frac{1}{N}\sum_{i=1}^{N}(y_i - \hat{y}_i)^2$$

A.11   ENTROPE MODEL ALGORITHMS

---

**Algorithm 1** EntroPE Forward Pass

---

**Require:** Raw time series $X \in \mathbb{R}^{B \times C \times L}$, forecast horizon $T$
**Ensure:** Predictions $\hat{Y} \in \mathbb{R}^{B \times C \times T}$
 1: $X^{ci} \leftarrow \text{reshape}(X, (B \cdot C, L))$
 2: $X^{norm}, \text{params} \leftarrow \text{RevIN\_forward}(X^{ci})$
 3: $X^{tok} \leftarrow \text{tokenize}(X^{norm})$
 4: $\mathcal{B}, M \leftarrow \text{EDP}(X^{tok}, \theta)$
 5: $P^{in}, H^{enc} \leftarrow \text{APE}(X^{norm}, \mathcal{B}, V)$
 6: $P^{out} \leftarrow \text{GlobalTransformer}(P^{in})$
 7: $H^{dec} \leftarrow \text{FusionDecoder}(P^{out}, H^{enc})$
 8: $\hat{Y}^{norm} \leftarrow W_{\text{proj}}(\text{flatten}(H^{dec}))$
 9: $\hat{Y}^{norm} \leftarrow \text{reshape}(\hat{Y}^{norm}, (B \cdot C, T))$
10: $\hat{Y} \leftarrow \text{RevIN\_inverse}(\hat{Y}^{norm}, \text{params})$
11: $\hat{Y} \leftarrow \text{reshape}(\hat{Y}, (B, C, T))$
12: **return** $\hat{Y}$ =0

---

---

**Algorithm 2** Entropy-Based Dynamic Patcher (EDP)

---

**Require:** Tokenized sequence $X^{tok} \in \mathbb{R}^{(B \cdot C) \times L}$, pre-trained entropy model $\theta$
**Ensure:** Patch boundaries $\mathcal{B}$, patch mask $M \in \mathbb{R}^{(B \cdot C) \times L}$
 1: Initialize $\mathcal{B} \leftarrow \emptyset$, $M \leftarrow \mathbf{0}$
 2: **Phase 1: Entropy Calculation**
 3: **for** $t = 1$ to $L - 1$ **do**
 4:    context $\leftarrow X^{tok}[:, 1:t]$
 5:    logits $\leftarrow \text{entropy\_model}_\theta(\text{context})$
 6:    $p_\theta(x_{t+1}|x_{\leq t}) \leftarrow \text{softmax}(\text{logits})$
 7:    $H(x_t) \leftarrow -\sum_{v \in \mathcal{V}} p_\theta(v) \log p_\theta(v)$
 8: **end for**
 9: **Phase 2: Boundary Detection**
10: **for** $t = 2$ to $L - 1$ **do**
11:    **if** $H(x_t) > \theta$ **and** $H(x_t) - H(x_{t-1}) > \gamma$ **then**
12:      $\mathcal{B} \leftarrow \mathcal{B} \cup \{t\}$, $M[:, t] \leftarrow 1$
13:    **end if**
14: **end for**
15: **Add first and last positions as boundaries**
16: $\mathcal{B} \leftarrow \{1\} \cup \mathcal{B} \cup \{L\}$
17: $M[:, 1] \leftarrow 1$, $M[:, L] \leftarrow 1$
18: **return** $\mathcal{B}, M$ =0

---

---

**Algorithm 3** Adaptive Patch Encoder (APE)

---

**Require:** Continuous sequence $X^{ci} \in \mathbb{R}^{(B \cdot C) \times L}$, patch boundaries $\mathcal{B}$, vocab size $V$
**Ensure:** Patch embeddings $P^{in}$, hidden states $H^{enc}$
 1: $E \leftarrow \text{EmbeddingLayer}(V, d_t)$
 2: $h_0 \leftarrow E(\text{tokenize}(X^{ci}))$                                              {Time-point embeddings}
 3: **Extract variable-length patches**
 4: patches $\leftarrow \{\}$
 5: **for** $i = 1$ to $|\mathcal{B}| - 1$ **do**
 6:     $\text{patch}_i \leftarrow X^{ci}[:, \mathcal{B}[i] : \mathcal{B}[i+1]]$
 7:     Append $\text{patch}_i$ to patches
 8: **end for**
 9: **Initial patch embedding via max pooling**
10: **for** each patch $p_j$ **do**
11:     $\text{emb} \leftarrow E(\text{tokenize}(p_j))$
12:     $P_0[:, j, :] \leftarrow \text{MaxPool}(\text{emb})$
13: **end for**
14: **Cross-attention refinement ($N$ layers)**
15: $h_{\text{curr}} \leftarrow h_0, \quad P_{\text{curr}} \leftarrow P_0$
16: **for** $n = 1$ to $N$ **do**
17:     $h_n \leftarrow \text{TransformerEncoder}_n(h_{\text{curr}})$
18:     $Q \leftarrow W_q(P_{\text{curr}})$
19:     $K \leftarrow W_k(h_n), \quad V \leftarrow W_v(h_n)$
20:     $\text{out} \leftarrow \text{softmax}\left(\frac{QK^T}{\sqrt{d_k}}\right) V$
21:     $P_n \leftarrow P_{\text{curr}} + W_o(\text{out})$
22:     Update $h_{\text{curr}}, P_{\text{curr}}$
23: **end for**
24: **return** $P^{in} \leftarrow P_N, \quad H^{enc} \leftarrow h_N =0$

---

