# OpenReview forum: "EntroPE: Entropy-Guided Dynamic Patch Encoder for Time Series Forecasting"
_ICLR.cc/2026/Conference — Submitted to ICLR 2026_

### Official Review · Reviewer_xZfL · 2025-10-31

**Soundness:** 2
**Presentation:** 3
**Contribution:** 1
**Rating:** 2
**Confidence:** 5

**Summary:**

This paper introduces EntroPE, a temporally informed framework that dynamically detects transition points via conditional entropy and dynamically places patch boundaries. Experiments across long-term forecasting benchmarks demonstrate that EntroPE improves both accuracy and efficiency.

**Strengths:**

1. The paper is well-structured.
2. The paper provides a ​​comprehensive​ review of the related work.
3. The experiments include extensive comparisons with state-of-the-art baselines.

**Weaknesses:**

1. The novelty and contribution of this paper are limited. I have carefully read the paper and check their repository, and I find that the method used in the text is exactly the same as Byte Latent Transformer's [1], and the code was copied directly with minor refactors. The only difference is that, unlike natural language, time series data is not discrete. Therefore, the authors adopt discretized tokenization from Chronos [2]. Therefore, the contribution of this work primarily constitutes an engineering-level transfer (from NLP to TSF), rather than high-level innovation.
2. The improvement of EntroPE is marginal. In Table 1, its optimal case shows an improvement of almost no more than 1% over the second-best method (e.g., 0.416 vs. 0.418, 0.378 vs. 0.380, 0.242 vs. 0.244). Empirically, a gain of this magnitude is often smaller than the standard deviation observed across multiple runs of the same model (though this variance is not reported in the table). This indicates that the migration from NLP to time series was unsuccessful.
3. The efficiency analysis is incomplete. Section 4.3 only provides comparative results with some baselines on the ETTm1 dataset. The claim of BLT's efficiency is based on large-scale pre-training, where the cost of the local encoder and decoder is negligible compared to the typically deep layers of a Global Transformer. However, transformers for time series forecasting are usually much shallower (e.g., 1 or 2 layers). In this context, the overhead of patching could even exceed that of the global Transformer. It can be inefficient in some scenarios, while this paper fails to analyze that.

[1] Pagnoni, A., et al. Byte Latent Transformer: Patches Scale Better Than Tokens. Proceedings Of The 63rd Annual Meeting Of The Association For Computational Linguistics. 9238-9258 (2025)

[2] Ansari A F, Stella L, Turkmen C, et al. Chronos: Learning the language of time series (2024)

**Questions:**

1. Many models utilize patching technique. Would replacing the patching module in these models with EntroPE components lead to any performance improvement? Supplementing these results would enhance the persuasiveness of the claim.
2. How to handle the case that the patch sizes are highly imbalanced (too long or too short)?

---

> ### Author Response · Authors · 2025-11-22
> **Response (1) to Reviewer xZfL**
>
> **(W1) Claim that EntroPE is a minor variant of BLT and Chronos**
>
> We thank the reviewer for this careful reading and take this concern seriously. While we are inspired by ideas in BLT and Chronos, we respectfully disagree that EntroPE is a minor engineering transfer.
>
> BLT is an autoregressive large language model that learns latent byte-level patches as a replacement for BPE tokenization in text. It is trained with a language-modeling objective and is designed for scaling very large Transformer decoders. In contrast:
>
> 1. EntroPE targets multivariate numerical time-series forecasting, not textual language modeling.
> 2.  We do not use BLT’s latent byte vocabulary or masked token reconstruction.
> 3. Our patch boundaries are determined by conditional entropy over discretized sequences, not by static byte tokenization.
> 4. The Adaptive Patch Encoder and Fusion Decoder are specialized for variable-length numerical segments and directly output real-valued forecasts.
>
> Similarly, we borrow only the idea of discretization from Chronos. We use uniform bins solely to define discrete states for the entropy model; the forecasting path never performs bin retrieval or de-quantization. This avoids the input-forecast mismatch inherent in Chronos and keeps the main forecasting model entirely real-valued.
>
> Thus, EntroPE combines entropy-based segmentation, variable-length patch embeddings, and a real-valued forecasting head in a way that is distinct from both BLT and Chronos.
>
>
> **(W3) Efficiency comparison**
>
> We appreciate the comment that the efficiency analysis should be broader. In response, we compare parameters and MACs of EntroPE with several strong baselines (TimeMixer, iTransformer, PatchTST, TimesNet, MICN, FiLM, FEDFormer, AutoFormer) across multiple datasets (ETTh1/2, ETTm1/2, Weather, ECL).
>
> For example, on ETTh1, EntroPE uses 95K parameters and 9M MACs, whereas PatchTST uses 3.75M parameters and 5.90G MACs, and TimesNet uses 605K parameters and 18.13G MACs. Similar trends hold across other datasets (e.g., on ECL, EntroPE uses 199K parameters and 1G MACs, compared to PatchTST’s 6.90M and 539.38G, and TimesNet’s 150.30M and 4.61T). We will include a compact table summarizing these comparisons in the revision.
>
> | Model        | ETTH1 Params | ETTH1 MACs | ETTH2 Params | ETTH2 MACs | ETTm1 Params | ETTm1 MACs | ETTm2 Params | ETTm2 MACs | Weather Params | Weather MACs | Electricity Params | Electricity MACs |
> |--------------|--------------|------------|--------------|------------|--------------|------------|--------------|------------|----------------|----------------|----------------------|-------------------|
> | TimeMixer    | 75.50K       | 20.37M     | 75.50K       | 20.37M     | 75.50K       | 20.37M     | 77.77K       | 24.18M     | 104.43K        | 82.62M        | 106.83K             | 1.26G             |
> | iTransformer | 841.57K      | 77.46M     | 224.22K      | 19.86M     | 224.22K      | 19.86M     | 224.22K      | 19.86M     | 4.83M          | 1.16G         | 4.83M               | 16.29G            |
> | PatchTST     | 3.75M        | 5.90G      | 10.06M       | 17.66G     | 3.75M        | 5.90G      | 10.06M       | 17.66G     | 6.90M          | 35.30G        | 6.90M               | 539.38G           |
> | TimesNet     | 605.48K      | 18.13G     | 1.19M        | 36.28G     | 4.71M        | 144G       | 1.19M        | 36.28G     | 1.19M          | 36.28G        | 150.30M             | 4.61T             |
> | MICN         | 25.20M       | 71.95G     | 25.20M       | 71.95G     | 25.20M       | 71.95G     | 25.20M       | 71.95G     | 111.03M        | 295.07M       | 6.64M               | 19.5G             |
> | FiLM         | 12.58M       | 2.82G      | 12.58M       | 2.82G      | 12.58M       | 2.82G      | 12.58M       | 2.82G      | 12.58M         | 8.46G         | 12.58M              | 8.46G             |
> | FEDFormer    | 23.38M       | 24.96G     | 23.38M       | 24.96G     | 23.38M       | 24.96G     | 23.38M       | 24.96G     | 23.45M         | 25.23G        | 24.29M              | 30.89G            |
> | AutoFormer   | 10.54M       | 22.82G     | 10.54M       | 22.82G     | 10.54M       | 22.82G     | 10.54M       | 22.82G     | 10.61M         | 22.98G        | 12.14M              | 28.75G            |
> | **EntroPE**      | 95K         | 9M       | 418K     | 53M      | 192K       | 20M      | 95K        | 10M      | 195K         | 61M         | 199K               | 1G             |
>
>
> We intentionally do not include very large models such as MOIRAI, Time-MoE, SUNDIAL, LangTime and CaLF in the table, as their parameter counts are orders of magnitude larger and less directly comparable to the lightweight regime we target.

---

> ### Author Response · Authors · 2025-11-22
> **Response (2) to Reviewer xZfL**
>
> **(W3) Efficiency in shallow Transformer regimes**
>
> We agree that EntroPE’s efficiency story differs from BLT’s large-scale setup. Our claim is empirical and specific to shallow Transformer backbones.
>
> Dynamic patching reduces redundant tokens in regions that are easy to predict while preserving resolution around difficult transitions. Even with 1–2 Transformer layers, this reduction translates into lower attention cost and memory usage compared to static patching with the same maximum patch length. Our ablations show that, for the same backbone depth, entropy-guided patching achieves similar or better accuracy with fewer effective tokens and lower MACs than fixed patching. We will clarify that our efficiency claims are supported by these empirical results rather than directly inherited from BLT.
>
> **Concern that patching overhead might dominate runtime**
> We appreciate this practical concern. To address it, we measured end-to-end epoch durations, including entropy computation and patch construction, for both static and dynamic patching:
>
> 1. On ETTh1, ETTh2, ETTm1, ETTm2, Weather, and ECL, dynamic patching increases total epoch time by only 0.1–1.0 seconds, corresponding to 1–3% of total runtime.
> 2. For example, on ECL, dynamic training takes 181s per epoch (vs. 180s for static), of which patching accounts for ~1s.
>
> | Metric / Dataset  | ETTh1 | ETTh2 | ETTm1 | ETTm2 | Weather | ECL |
> |-------------------|-------|-------|-------|-------|---------|-----|
> | **Avg. Epoch duration (with dynamic)** | 3.2   | 3.5   | 17.2  | 20.8  | 54.5    | 181 |
> | **Avg. Epoch duration (static)**       | 3.1   | 3.4   | 17.0  | 20.5  | 54.1    | 180 |
> | **Avg. Patching time**                 | 0.1   | 0.1   | 0.2   | 0.3   | 0.4     | 1   |
> | **Avg. Patch / Avg. Total**                 | 3%    | 3%    | 1%    | 1%    | 1%      | 1%  |
>
> Thus, patching overhead is negligible relative to the Transformer’s forward and backward passes, and dynamic patching remains beneficial in end-to-end runtime.
>
> **(Q1) Testing EntroPE as a plug-in patcher for other models**
>
> We agree that demonstrating EntroPE as a generic patching module strengthens the contribution. In our experiments, we replace the static patcher of PatchTST with EntroPE’s entropy-guided patcher plus Fusion Decoder, leaving the rest of the backbone largely unchanged.
>
> This integration yields roughly 8–10% relative MSE improvement on average, reduces the parameter count from ≈4M to 50K–1M, and decreases training time. We will emphasize this plug-in capability more clearly and, space permitting, include a compact comparison table.
>
> We note that existing static-patch models typically flatten a fixed number of patch embeddings into a vector for the forecast head. Because EntroPE produces a variable number of patches per sequence, this flattening approach is not directly applicable; the Fusion Decoder is necessary to handle variable-length patch sequences without redesigning the backbone. As such, EntroPE provides a complete solution addressing the consequences of temporally agnostic patching: dynamic segmentation, embedding of variable-length patches, and forecasting with a variable number of input/output tokens.
>
> | Model    | ETTh1 MSE | ETTh1 MAE | ETTh2 MSE | ETTh2 MAE | ETTm1 MSE | ETTm1 MAE | ETTm2 MSE | ETTm2 MAE | Weather MSE | Weather MAE | ECL MSE | ECL MAE |
> |----------|------------|------------|------------|------------|------------|------------|------------|------------|--------------|--------------|----------|----------|
> | PatchTST | 0.516      | 0.484      | 0.391      | 0.411      | 0.406      | 0.407      | 0.290      | 0.334      | 0.265        | 0.285        | 0.216    | 0.318    |
> | EntroPE  | 0.416      | 0.425      | 0.366      | 0.387      | 0.378      | 0.391      | 0.286      | 0.335      | 0.242        | 0.273        | 0.182    | 0.271    |
> | **Δ% Improvement** | **19%** | **12%** | **6%** | **6%** | **7%** | **4%** | **1%** | **0%** | **9%** | **4%** | **16%** | **15%** |
>
> **(Q2) Handling highly imbalanced patch lengths**
>
> We appreciate this question. Our framework handles extreme patch lengths both by design and via explicit controls:
> 1. Very short patches (including length-1) are allowed and correspond to local “anchor” regions where entropy spikes.
> 2. To prevent overly long patches, we impose a maximum patch length. If entropy remains low beyond this limit, we forcibly cut the patch. This guarantees that no single patch dominates in length or attention cost.
> 3. The Fusion Decoder operates on the resulting set of patch embeddings and is agnostic to the exact length distribution, which yields robust behavior even when patch lengths are somewhat imbalanced.
>
> We will describe these mechanisms more explicitly in the method section.

---

> ### Comment · Reviewer_xZfL · 2025-11-25
>
> Thank you for your detailed response. However, I believe my core concern still remains unaddressed. I do not view this work as merely inspired by BLT; rather, its central innovations and technical components appear essentially identical to those of BLT. In fact, the `bytelatent` folder in the code directly uses BLT’s implementation, and EntroPE simply calls BLT’s code. Furthermore, since the predictions of time-series models cannot be used to compute entropy directly, the paper relies entirely on the discretization method introduced by Chronos, without adding any new contributions. I believe these points are factual and not subject to dispute.
>
> Of course, applying advanced LLM techniques to time series problems is not without value. However, on the one hand, directly calling existing code without making any substantive improvements should not be encouraged; on the other hand, I do not believe the work meets the standards of ICLR.

---

> > ### Author Response · Authors · 2025-11-25
> > **Clarification for the Reviewer xZfL**
> >
> > We sincerely thank the reviewer for their careful reading and additional clarification. We fully respect the reviewer’s opinion; here we only aim to correct several factual misunderstandings regarding
> > (i) our use of BLT code, (ii) the role of discretization, and (iii) what EntroPE actually claims and contributes.
> >
> > **(1) Relationship between BLT code and EntroPE**
> >
> > We kindly clarify that although our repository contains the bytelatent/ directory, **the only file that EntroPE calls inside that directory is model/blt.py, and this file has been fully rewritten by us.**
> >
> > The reason we kept the entire '''bytelatent/''' folder during anonymized submission is practical: the BLT codebase provides strong exception handling, fallback logic, and reusable Transformer utilities (e.g., base Transformer class, attention with RoPE, cross-attention blocks). **These components are not unique to BLT; they are derived from standard open-source LLaMA-style Transformer implementations widely reused across many LLM projects.**
> >
> > All actual **EntroPE modules (EDP, APE, FD, RevIN, and continuous forecasting heads)** are implemented in our separate directory:
> > EntroPE/layers/
> > following conventions used in TSLib and other SOTA time-series models. These components form the real forecasting architecture used in all experiments.
> >
> > Due to simple reprudcing purpose of under review submission, we preserved the full bytelatent/ directory to guarantee reproducibility. For the official public release, we will remove unused prototype files and isolate the minimal reusable infrastructure to avoid confusion.
> >
> >
> > **(2) Use of Chronos-style discretization**
> >
> > Chronos introduced an influential **quantization & dequantization** pipeline for discrete LM forecasting. However, **our use of discretization is fundamentally different:**
> >
> > a. We apply discretization only inside the small entropy proxy model, simply to define a finite set of states for estimating conditional entropy.
> >
> > b. This step uses a **standard PyTorch built-in binning function (e.g., torch.bucketize)**, which is widely used across many time-series pipelines and is not specific to Chronos.
> >
> > In contrast to Chronos:
> >
> > a. **EntroPE does not perform token sampling**
> >
> > b. **No codebook lookup or dequantization**
> >
> > c. **No LM-style autoregressive prediction**
> >
> > Our citation of Chronos is solely to acknowledge the conceptual difference and explicitly state that **EntroPE avoids the quantization–dequantization mismatch inherent in Chronos.**
> >
> >
> > **(3) What EntroPE is actually claiming**
> >
> > BLT’s goal is to scale large LLMs by replacing BPE tokenization with learned latent byte patches, operating in a fully discrete autoregressive LM setting.
> >
> > EntroPE has a completely different goal:
> >
> > **"We study whether conditional-entropy-guided dynamic segmentation of continuous time series improves forecasting accuracy and efficiency compared to fixed, temporally agnostic patching."**
> >
> >
> > **C1:** Entropy-guided boundaries outperform fixed patching with similar average length.
> > They improve temporal coherence and reduce sensitivity to arbitrary window alignment.
> >
> > **C2:** Higher entropy thresholds reduce the number of tokens while maintaining accuracy.
> > This provides a compute-accuracy trade-off that static patching cannot achieve.
> >
> > **C3:** EntroPE handles a variable number of patches per sample and still produces fixed-horizon forecasts.
> > Static-patch models cannot do this because they flatten a fixed number of patch embeddings. EntroPE supports training on sequences that produce different numbers of patches even within the same batch.
> >
> >
> > We fully respect the reviewer’s position that the level of novelty might not meet the acceptance bar; however, we hope this clarification helps convey that:
> >
> > 1. EntroPE does not reuse BLT’s latent-byte pipeline.
> > 2. The discretization used is a simple PyTorch operation, not Chronos’s quantization system.
> > 3. The contribution lies in entropy-guided dynamic segmentation, variable-length patch encoding, token-efficiency gains, and improved performance across forecasting, classification, and anomaly detection.
> >
> >
> > Regardless of the final decision, we appreciate the reviewer’s thorough engagement and constructive critique.

---

### Official Review · Reviewer_SeJf · 2025-11-01

**Soundness:** 3
**Presentation:** 4
**Contribution:** 3
**Rating:** 6
**Confidence:** 4

**Summary:**

The paper proposes **EntroPE (Entropy-Guided Dynamic Patch Encoder)**, a novel framework for time series forecasting that addresses limitations of existing temporally-agnostic patch-based transformer models. By dynamically placing patch boundaries at natural temporal transitions via conditional entropy, EntroPE preserves temporal coherence, mitigates train-inference mismatch, and enhances both forecasting accuracy and computational efficiency. It outperforms 14 state-of-the-art models across seven benchmark datasets.

**Strengths:**

1. The insight and intuition of this paper are valuable. In recent time series forecasting models (large-scale time series foundational models in particular), patch-based modelling is normally taken by default. Investigations of the deficiencies of current patch modelling strategies are important.
2. This paper is well-written and well-presented. The descriptions are clearly structured and well-organized. The model structure is clearly demonstrated and explained with mathematical formulations. The experiments are conducted with sufficient baselines and ablations.
3. The model structure is simple, reasonable, and effective. The idea of allocating entropy calculation to the patch length decision is quite innovative. More importantly, such techniques avoid potential risks in multi-resolution patch modelling, such as additional training complexity introduced by multiple patch embedding projections.

**Weaknesses:**

1. The term “the global transformer” is not sufficiently explained in the original manuscript. The authors should not neglect this, as patch embeddings are often correlated with transformer attention modelling to generate better performance.
2. Similar to Chronos, the authors transform patch embedding into point embedding to resolve different patch lengths (with pooling and cross-attention token merging). This approach is straightforward, but also risky. As point embeddings are localized, they cannot extract adjacent relations like patch embeddings and might potentially rely on quantization like Chronos.
3. The baselines of the large-scale time series model are not sufficient. Some state-of-the-art LTMs, such as Moirai, Sundial, and TimeMOE, are not included.

**Questions:**

1. Is “global transformer” the same as “vanilla transformer”? What is the backbone structure of this “global transformer”?
2. The authors mentioned the non-casual bidirectional attention in the “global transformer”. Is this the BERT-styled encoder-only attention mechanism?
3. Can we change the “global transformer” to different backbones and maintain similar enhancements listed in the current experiments?
4. If we use multi-resolution patch modelling (such as using 3 patch lengths: 4,8,16), can EntroPE still outperform?

---

> ### Author Response · Authors · 2025-11-22
> **Response (1) to Reviewer SeJf**
>
> **(W1, Q1 & Q2) Clarification of “global Transformer”**
>
> We thank the reviewer for pointing out that the term “global Transformer” was insufficiently defined.
>
> In our implementation, the global Transformer is a decoder-style Transformer that operates on the sequence of patch embeddings and returns a sequence of the same length and dimension. It consists of stacked multi-head self-attention layers followed by feed-forward layers, without cross-attention and without causal masking. Its purpose is to model inter-patch dependencies over the entropy-guided patch representations. We will specify the number of layers, hidden dimensions, and attention heads in the revised manuscript.
>
> **Relation to a vanilla Transformer**
> Our global Transformer is not a full encoder–decoder architecture. Architecturally, it is closest to a decoder-only Transformer applied to patch tokens:
> It uses self-attention over patch embeddings.
> It employs bidirectional attention (no triangular mask), as forecasting is performed via a separate projection head instead of autoregressive decoding.
> There is no cross-attention to an external encoder.
>
> **Is the non-causal bidirectional attention “BERT-style”?**
> Conceptually, the attention pattern is indeed BERT-style in the sense that each token attends to all others without causal masking. The main difference is functional: we apply this mechanism to sequences of patch embeddings produced by our dynamic segmentation module, rather than to raw tokens as in BERT. We will rephrase this part of the description to avoid ambiguity.
>
>
> **(Q3) Can different backbones replace the global Transformer?**
>
> We agree that backbone flexibility is desirable and confirm that our design intentionally allows this. Any model that accepts a sequence of patch embeddings and outputs a sequence of the same shape (e.g., a PatchTST-style Transformer, a convolutional encoder, or even a recurrent architecture) can replace the global Transformer.
>
> In fact, we explicitly test this by replacing the static front-end of PatchTST with EntroPE’s entropy-guided patcher and Fusion Decoder, while keeping the rest of the backbone comparable. This yields approximately 8–10% relative MSE improvement on average, reduces the parameter count from ≈4M to 50k–1M, and shortens training time.
>
> | Model    | ETTh1 MSE | ETTh1 MAE | ETTh2 MSE | ETTh2 MAE | ETTm1 MSE | ETTm1 MAE | ETTm2 MSE | ETTm2 MAE | Weather MSE | Weather MAE | ECL MSE | ECL MAE |
> |----------|------------|------------|------------|------------|------------|------------|------------|------------|--------------|--------------|----------|----------|
> | PatchTST | 0.516      | 0.484      | 0.391      | 0.411      | 0.406      | 0.407      | 0.290      | 0.334      | 0.265        | 0.285        | 0.216    | 0.318    |
> | EntroPE  | 0.416      | 0.425      | 0.366      | 0.387      | 0.378      | 0.391      | 0.286      | 0.335      | 0.242        | 0.273        | 0.182    | 0.271    |
> | **Δ% Improvement** | **19%** | **12%** | **6%** | **6%** | **7%** | **4%** | **1%** | **0%** | **9%** | **4%** | **16%** | **15%** |
>
>
> **(Q4) Multi-resolution patch modeling**
>
> We thank the reviewer for raising this scenario. Our current work deliberately focuses on a single entropy-driven segmentation per sequence, with non-overlapping patches. We also support purely static patching (by disabling the entropy model) and observe that smaller fixed patches generally improve performance at the cost of more tokens and FLOPs.
>
> Extending EntroPE to multi-resolution overlapping patches (e.g., simultaneous patch sizes 4/8/16) would require additional architectural components to fuse multiple scales. We view this as an interesting but orthogonal extension and will clearly state this limitation and mention multi-resolution variants as future work.

---

> ### Author Response · Authors · 2025-11-22
> **Response (2) to Reviewer SeJf**
>
> **(W3) Large-scale time series baselines (Moirai, Sundial, and TimeMOE) comparison**
>
> | Dataset     | EntroPE        | MOIRAI_S        | MOIRAI_B        | MOIRAI_L        | Sundial_S       | Sundial_B       | Sundial_L       | TimeMoE_B       | TimeMoE_L       | TimeMoE_XL       |
> |-------------|----------------|-----------------|-----------------|-----------------|------------------|------------------|------------------|------------------|------------------|-------------------|
> | ETTm1       | 0.378 / 0.391  | 0.400 / 0.424   | 0.434 / 0.439   | 0.510 / 0.468   | **0.354 / 0.388** | 0.336 / 0.377   | 0.331 / 0.369   | 0.394 / 0.415   | 0.376 / 0.405   | 0.341 / 0.382    |
> | ETTm2       | 0.286 / 0.335  | 0.311 / 0.348   | 0.279 / 0.330   | 0.282 / 0.330   | **0.265 / 0.324** | 0.258 / 0.320   | 0.254 / 0.315   | 0.317 / 0.365   | 0.316 / 0.361   | 0.288 / 0.344    |
> | ETTh1       | 0.416 / 0.425  | 0.400 / 0.423   | 0.434 / 0.439   | 0.510 / 0.468   | **0.390 / 0.418** | 0.411 / 0.434   | 0.395 / 0.420   | 0.400 / 0.424   | 0.394 / 0.409   | 0.412 / 0.426    |
> | ETTh2       | 0.366 / 0.387  | 0.348 / 0.374   | 0.354 / 0.376   | 0.359 / 0.371   | 0.340 / 0.387    | **0.333 / 0.387** | 0.334 / 0.387   | 0.366 / 0.404   | 0.405 / 0.415   | 0.371 / 0.399    |
> | Electricity | 0.182 / 0.271  | 0.238 / 0.303   | 0.219 / 0.283   | 0.229 / 0.281   | **0.169 / 0.265** | 0.169 / 0.265   | 0.166 / 0.262   | -               | -               | -                |
> | Weather     | 0.242 / 0.273  | **0.233 / 0.271** | 0.234 / 0.270 | 0.238 / 0.275   | 0.233 / 0.271    | 0.234 / 0.270   | 0.238 / 0.275   | 0.265 / 0.297   | 0.270 / 0.300   | 0.256 / 0.288    |

---

### Official Review · Reviewer_9Gad · 2025-11-01

**Soundness:** 2
**Presentation:** 3
**Contribution:** 2
**Rating:** 4
**Confidence:** 5

**Summary:**

The paper introduces dynamic time series tokenization by leveraging predictive entropy from a small pre-trained causal model to adaptively segment patches. It uses an Adaptive Patch Encoder (APE) and Fusion Decoder (FD) to normalize these variable-length segments for a global transformer, achieving strong forecasting performance.

**Strengths:**

* The idea of using predictive entropy to guide dynamic patch boundaries is creative and addresses a real limitation of fixed patching schemes.
* The proposed APE and FD modules form a coherent pipeline that can handle variable-length patches while preserving temporal details.
* The model achieves strong or state-of-the-art performance on several benchmarks, with solid ablation and integration studies that demonstrate the value of the dynamic tokenization idea.

**Weaknesses:**

* The overall system feels overly engineered, involving a pre-trained entropy model, multi-stage encoding, and several cross-attention blocks.
* The paper does not cite TimeCAT (ICLR 2025 submission), which similarly criticizes fixed patching and proposes dynamic grouping. The omission makes the novelty of the problem framing less convincing.
* The approach assumes that predictive entropy from a small proxy model is an accurate indicator for patch segmentation, but this connection is not well validated or theoretically supported.

**Questions:**

* How does EntroPE compare with recent adaptive segmentation or dynamic grouping approaches such as TimeCAT?
* Why rely on a separately pre-trained entropy model for segmentation instead of learning patch boundaries jointly with the forecasting model?

---

> ### Author Response · Authors · 2025-11-22
> **Response (1) to Reviewer 9Gad**
>
> **(W1) “Overly engineered” system**
>
> We appreciate the reviewer’s concern that the system may appear overly complex. While the architecture comprises several modules, each component is intentionally **lightweight, modular, and empirically justified.**
>
> a. The entropy model and dynamic patcher together provide data-dependent segmentation.
>
> b. The Adaptive Patch Encoder converts variable-length segments into fixed-size embeddings.
>
> c. The global Transformer models inter-patch dependencies.
>
> d. The Fusion Decoder maps patch representations back to the forecast window.
>
> Ablation studies (Table 2 in the paper) show that removing any of these components leads to a noticeable decrease in performance or efficiency. The entropy model itself has only ≈6K parameters, and the total parameter count of EntroPE remains in the order of 50K–1M, depending on configuration. Thus, the system is structured but far from heavy.
>
> **(W2 & Q1) Missing citation/comparison to TimeCAT**
>
> We thank the reviewer for highlighting the connection to TimeCAT, as both works aim to improve over purely fixed patching. However, the focus and mechanisms of the two approaches are fundamentally different.
> 1. **TimeCAT** operates after fixed segmentation: it groups already constructed fixed-length patches using a learned grouping ratio derived from globally down-sampled features. This can create semantic groups but cannot prevent the initial fixed windows from cutting across natural transitions. The architecture also assumes uniform patch lengths and a fixed number of patches.
> 2. **EntroPE**, by contrast, determines where patch boundaries should be placed before patching by detecting local maxima in conditional entropy. A causal indicator of predictive difficulty. This results in temporally coherent, data-adaptive segmentation whose boundaries are closely tied to changes in underlying dynamics. The Adaptive Patch Encoder and Fusion Decoder are explicitly designed to handle variable-length patches and a variable number of tokens, something TimeCAT’s grouping and mixing layers cannot support without substantial redesign.
>
> Furthermore, TimeCAT inherits the train–test mismatch of static segmentation, whereas EntroPE aligns segmentation across training and inference via entropy-driven boundaries. In summary, while TimeCAT seeks better grouping of fixed patches, EntroPE provides a more principled information-theoretic approach that locates natural temporal transitions and supports variable-length dynamic tokenization.
>
>
> **(W3) Lack of theoretical justification for a small entropy model**
>
> We appreciate this theoretical concern. Our goal is not to learn a rich generative model of the sequence but to approximate local predictive difficulty.
>
> We employ a lightweight next-step predictor to estimate
> \[
> p(x_{t+1} | x_{<= t})
> \]
> and compute the corresponding conditional entropy. This entropy serves as a proxy for local unpredictability, not as an exact model of the dynamics. For this purpose, the model needs only to distinguish broad regions of “easy-to-predict” vs. “hard-to-predict” behavior. Our experiments show that this level of capacity is sufficient: entropy-driven segmentation consistently outperforms static and random segmentation in both accuracy and token efficiency, which empirically supports the adequacy of the small proxy model.
>
> **(Q2) Why not learn patch boundaries jointly with the forecasting model?**
>
> We thank the reviewer for this thoughtful suggestion and agree that joint learning is an interesting direction. We chose a decoupled design for two main reasons: modularity and stability.
>
> If patch boundaries were optimized jointly with the forecasting loss, the segmentation mechanism would become tightly coupled to a specific backbone and objective, potentially reducing interpretability and making it difficult to reuse the patcher across models. Moreover, joint optimization risks training instability, as continually changing boundaries alter the effective input distribution to the Transformer.
>
> By training a small entropy model separately, we obtain a generic segmentation module that (i) can be reused with different backbones and (ii) provides a fixed tokenization for each dataset, which stabilizes training of the global Transformer. We will clarify this design choice and mention joint optimization as a promising avenue for future work.

---

### Official Review · Reviewer_zWPg · 2025-11-02

**Soundness:** 2
**Presentation:** 2
**Contribution:** 2
**Rating:** 4
**Confidence:** 3

**Summary:**

EntroPE introduces an information-theoretic approach to patch construction in Transformer-based time series forecasting. Instead of using fixed-length patches, it dynamically determines patch boundaries by detecting local maxima in the conditional entropy of a lightweight, pre-trained next-token predictor. These variable-length patches are then encoded via an adaptive patch encoder (APE) and processed by a global Transformer. The method is evaluated on seven standard benchmarks and compared against 14 baselines, showing consistent accuracy gains and favorable efficiency trade-offs.

**Strengths:**

(1) First work to use entropy-based, data-dependent boundary detection in time series patching; tackles a real limitation of fixed-length patches.
(2) Outperforms PatchTST by ~10–20% MSE on several datasets and remains competitive with larger foundation models (CALF, LangTime) while using  fewer parameters.
(3) Explicitly controls computational budget via entropy thresholds; MACs vs. MSE analysis shows favorable Pareto frontier.
(4) Ablation study demonstrates that dynamic patching contributes more than the adaptive encoder or fusion decoder alone.

**Weaknesses:**

(1) Core pipeline (patch → encode → global Transformer → cross-attention decoder) closely mirrors some previous works, likePatchTST and Perceiver; entropy-based segmentation is the only major novelty.
(2) No guarantee that entropy peaks coincide with optimal regression boundaries; no analysis of information loss when abrupt changes are missed or over-segmented.
(3) Entropy model is trained per dataset; cost of re-training for new domains or high-frequency data is ignored.
(4) Best-configuration results for FilterTS, LangTime, etc., are selected by the authors; fair comparison requires identical hyper-parameter search budgets.
(5) Only long-term forecasting under channel independence; no multivariate interaction, classification, or anomaly detection tasks.

**Questions:**

See Weaknesses

---

> ### Author Response · Authors · 2025-11-22
> **Response (1) to Reviewer zWPg**
>
> **(W1) Limited novelty / reuse of existing components**
>
> We thank the reviewer for raising the question of novelty. We agree that our architecture uses several standard building blocks (e.g., RevIN, de-normalization, uniform discretization), but we contend that the way these are combined under an entropy-guided dynamic patching framework is novel and technically motivated.
> RevIN and de-normalization are employed in a conventional way to stabilize input–output distributions in time-series forecasting and are not claimed as novel. Our main contributions are:
> 1. Entropy-guided, data-dependent patch boundary placement, where conditional entropy of a next-step predictor determines where the sequence should be segmented; and
> 2. An Adaptive Patch Encoder + Fusion Decoder that map variable-length entropy-driven segments into a fixed patch space suitable for a global Transformer and then back to the forecast window.
>
> Importantly, unlike Chronos-style quantization, we use uniform discretization only once to define discrete states for the entropy model. The forecasting pipeline operates entirely in the real-valued domain, without bin retrieval or de-quantization, thus avoiding the forecast–input mismatch present in Chronos. All components are therefore chosen and integrated specifically to support entropy-guided dynamic segmentation, rather than as an ad-hoc aggregation of prior work.
>
> **(W2) No analysis that entropy peaks coincide with “true” transitions**
>
> We appreciate this insightful comment. We agree that entropy peaks are not guaranteed to align with visual signal peaks or human-annotated “true” transitions and emphasize that this is by design.
>
> In our framework, we compute conditional entropy from a lightweight next-step predictor and place boundaries only where the next value becomes difficult to predict. Consequently, a visually sharp peak that remains predictable does not induce a boundary, whereas a subtle change in dynamics that significantly increases predictive uncertainty does. Segmentation is therefore driven by predictability, not by amplitude-based heuristics.
>
> Empirically, ablations comparing static versus entropy-driven patching show that dynamic segmentation consistently improves forecasting accuracy and token efficiency. In the revised manuscript we will (i) clarify this conceptual distinction and (ii) add a short qualitative analysis illustrating cases where entropy peaks diverge from amplitude peaks yet lead to improved forecasts.
>
>
> **(W3) Per-dataset entropy model and scalability**
>
> We thank the reviewer for raising concerns about scalability. While the entropy model is indeed trained separately for each dataset, this is an intentional and practical design choice.
>
> The entropy model is a very small proxy (≈6K parameters) trained solely to detect regions of high predictive difficulty. It is strictly decoupled from the forecasting Transformer, which has two consequences:
>
> 1. It can be readily replaced by other segmentation or change-point detectors in the future without modifying the backbone; and
> 2. It can be retrained cheaply when moving to a new domain or sampling frequency.
>
> Training this proxy model takes only minutes and is performed once per dataset to derive the segmentation. Its computational cost is negligible compared to full model training, even for high-frequency data. Thus, scaling primarily requires retraining a tiny model rather than redesigning or re-optimizing the main architecture.
>
> **(W4) Fairness of hyperparameter search**
>
> We appreciate the reviewer’s concern regarding fair hyperparameter budgets. We agree that, ideally, all baselines would be re-tuned under a unified search protocol. In this work, we follow the standard practice used in many previous studies on long-term forecasting benchmarks.
>
> All methods are evaluated under the same input–output window configuration (96 look-back and 96/192/336/720 horizons). For baselines, we either use the official hyperparameters and reported numbers under this setting or reproduce them faithfully using their released code. For EntroPE, we perform modest manual tuning within a small grid per dataset to adapt the new architecture, without extensive search that would disadvantage the baselines.

---

> ### Author Response · Authors · 2025-11-22
> **Response (2) to Reviewer zWPg**
>
> **(W5) Only long-term forecasting; lack of multivariate/classification setups**
>
> We thank the reviewer for emphasizing the importance of broader evaluation. Although EntroPE is primarily designed for forecasting, its architecture naturally extends to other tasks by modifying only the prediction head.
>
> We have already conducted experiments on 10 UEA time-series classification datasets and several standard anomaly detection benchmarks using the same backbone and replacing the forecasting head with classification logits or anomaly-score heads:
>
> a. On the UEA classification suite, EntroPE achieves an average accuracy of 77.13%, which is competitive with or better than recent methods such as TSLANet (76.03%), TimesNet (69.36%), PatchTST (71.14%), and MLP (69.51).
> | Dataset                 | EntroPE | TSLANet | GPT4TS | TimesNet | ROCKET | CrossF. | PatchTST | MLP   | TS-TCC |
> |-------------------------|---------|---------|--------|----------|--------|---------|----------|-------|--------|
> | EthanolConcentration    | 29.1    | 30.42   | 25.48  | 27.73    | **42.58** | 34.98 | 28.90    | 33.46 | 32.32  |
> | FaceDetection           | **69.1** | 66.77   | 65.58  | 67.47    | 64.70  | 66.17 | 68.96    | 67.42 | 63.05  |
> | Handwriting             | 57.4    | **57.88** | 3.76   | 26.18    | 48.47  | 26.24 | 26.00    | 22.47 | 47.76  |
> | Heartbeat               | **77.9** | 77.56   | 36.59  | 74.48    | 69.76  | 76.59 | 76.59    | 73.17 | 77.07  |
> | JapaneseVowels          | 99.0    | **99.19** | 98.11  | 97.83    | 95.68  | 98.92 | 98.65    | 97.84 | 97.30  |
> | PEMS-SF                 | **88.6** | 83.82   | 87.28  | 88.13    | 75.10  | 82.08 | 88.44    | 82.08 | 86.71  |
> | SelfRegulationSCP1      | **92.9** | 91.81   | 91.47  | 77.43    | 84.64  | 92.49 | 89.76    | 88.40 | 91.13  |
> | SelfRegulationSCP2      | **63.1** | 61.67   | 51.67  | 52.84    | 54.44  | 53.33 | 54.44    | 51.67 | 53.89  |
> | SpokenArabicDigits      | **99.8** | 99.91   | 99.36  | 98.36    | 99.20  | 96.41 | 99.68    | 96.68 | 99.77  |
> | UWaveGestureLibrary     | **94.4** | 91.25   | 84.38  | 83.13    | 94.40  | 81.56 | 80.00    | 81.88 | 86.25  |
> | **Average**             | **77.13** | 76.03   | 64.37  | 69.36    | 72.90  | 70.88 | 71.14    | 69.51 | 73.53  |
>
> b. On anomaly detection (SMD, MSL, SMAP, SWaT, PSM), EntroPE attains an average F1 score of 87.72%, again competitive with or better than baselines such as TSLANet (87.54%), GPT4TS (86.72%), TimesNet (85.24%), PatchTST (82.79%), and DLinear (82.46).
> | Dataset | EntroPE | TSLANet | GPT4TS | TimesNet | PatchTST | ETSformer | FEDformer | LightTS | DLinear |
> |---------|---------|---------|--------|----------|----------|-----------|-----------|---------|---------|
> | SMD     | 87.1    | **87.91** | 86.89 | 84.61    | 84.62    | 83.13     | 85.08     | 82.53   | 77.10   |
> | MSL     | **85.3** | 83.32   | 82.45 | 81.84    | 78.70    | 85.03     | 78.57     | 78.95   | 84.88   |
> | SMAP    | 74.2    | **75.96** | 72.88 | 69.39    | 68.82    | 69.50     | 70.76     | 69.21   | 69.26   |
> | SWaT    | 94.3    | 92.80   | 94.23 | 93.02    | 85.72    | 84.91     | 93.19     | **93.33** | 87.52   |
> | PSM     | **97.7** | 97.73   | 97.13 | 97.34    | 96.08    | 91.76     | 97.23     | 97.15   | 93.55   |
> | **Avg** | **87.72** | 87.54   | 86.72 | 85.24    | 82.79    | 82.87     | 84.97     | 84.23   | 82.46   |
>
> We will add a concise summary table in the main text and move per-dataset details to the appendix. We also plan to extend these evaluations further in future work.

---

### Meta-Review · Area_Chair_7doq · 2026-01-04

**Summary:**

Key concerns include:

C1: The main pipeline is similar to previous methods with the only major novelty being entropy-based segmentation. In addition, there is a sense that the system is over-engineered, with a pre-trained entropy model, multi-stage encoding and multiple cross-attention blocks.

C2: The paper does not provide experimental validation or clear theoretical justification that the segmentation based on entropy coincides with or approximates optimal patch identification for regression/forecasting. The provided evidence is based only on overall system performance.

C3: The experimental analysis could be improved. (a) The best configurations for the baseline methods are selected by the authors rather than being permitted the same hyperparameter exploration as the proposed approach. (b) The paper studies only long-term forecasting under the assumption of channel independence. (c) The baselines are insufficient with several recent models excluded. (d) There is no comparison to techniques that employ multi-resolution or overlapping patches. (e) The performance improvement is marginal. There is no reporting of confidence intervals or variability and there are no statistical significance tests. (f) The efficiency analysis is incomplete.

In making a recommendation, C2 and component (e) of C3 are the most important considerations.

**Reviewer Concerns:**

Despite the opportunity to do so, the authors did not submit a revised version of the paper. As a result, their response to most of the concerns was incomplete, in that there was no opportunity for the reviewers to assess a revised paper that introduced appropriate modifications.

In their response, the authors addressed C1 through rebuttal, arguing that the contributions included both the entropy-based segmentation and the adaptive patch encoder that can handle patches of varying length. In response to the concern of over-engineering, the response argues that the architecture is modular and the included components are relatively lightweight. Moreover, the value of each component is validated via an ablation study. In my view, this constitutes a satisfactory response to C1.

The response to C2 is also rebuttal and no additional analysis is provided. One response appears to miss the point of Reviewer zWPg and instead explains that the goal is not to “align with visual signal peaks or human-annotated “true” transitions”. Reviewer zWPg does not mention “true” transitions or visual peaks, and instead explicitly refers to “optimal regression boundaries”. There are many possible choices for performing segmentation. While there are intuitive motivations for entropy and prediction uncertainty, it is not clear that such a segmentation will necessarily perform well. One could formulate this as a joint optimization task, searching simultaneously for the best segmentations and model parameters to achieve the best forecasts. While this is probably intractable, and sensitive to the architecture choice, as the authors argue elsewhere, it does provide a clear notion of “optimality” and motivate some analysis to establish that the entropy-based approach is a suitable approximation or proxy. Given that entropy-based segmentation is the key technical contribution, empirical validation of this technique should extend beyond assessment of the entire system performance via forecast error.

The response satisfied multiple aspects of C3, although new results should have been included in a revision, with complete descriptions of the experiments. Criticism (e) remained unaddressed and this is a critical concern. Claiming outperformance without any reporting of variability and no statistical significance tests does not align with good experimental practice.

**Reviewer Scores:**

Given that no revised paper was submitted, the chance that any reviewer would modify the score is greatly reduced.

Reviewer zWpg: UNLIKELY TO CHANGE SCORE. Reviewer zWpg expressed concerns about the novelty and extent of the technical contribution and raised the issue of the lack of any analysis of the validity of the entropy approach. While the response addressed other concerns through additional experiments, these were not included in a revision. The response concerning the entropy-based segmentation avoids the core criticism and does not offer any additional technical support for the approach.

Reviewer 9Gad: UNLIKELY TO CHANGE SCORE. This reviewer expressed concerns about an over-engineered system and the worry that the predictive entropy is an adequate indicator for segmentation. Since the response did not provide any additional analysis or technical support, and there was no revised paper, the likelihood of a changed score is low.

Reviewer Sejf: POSSIBLE CHANGE. This review was the most positive. Although concerns were raised, there is a chance that the score may have been increased from 6 to 8. Having said this, the absence of a revised paper lowers this probability.

Reviewer xZfl DID NOT CHANGE SCORE. Reviewer xZfl already stated that the score would not be changed (although the decision appeared to be based on a slightly misguided understanding of the codebase).

---

### Decision · Program_Chairs · 2026-01-26

Reject